# Structure of the malaria vaccine candidate antigen CyRPA and its complex with a parasite invasion inhibitory antibody

Paola Favuzza[1,2], Elena Guffart[3], Marco Tamborrini[1,2], Bianca Scherer[1,2], Anita M Dreyer[1,2], Arne C Rufer[3], Johannes Erny[3], Joerg Hoernschemeyer[3], Ralf Thoma[3], Georg Schmid[3], Bernard Gsell[3], Araceli Lamelas[1,2†], Joerg Benz[3], Catherine Joseph[3], Hugues Matile[3], Gerd Pluschke[1,2]*, Markus G Rudolph[3]*

[1]Medical Parasitology and Infection Biology Department, Swiss Tropical and Public Health Institute, Basel, Switzerland; [2]University of Basel, Basel, Switzerland; [3]Roche Pharmaceutical Research and Early Development, Small Molecule Research, Roche Innovation Center Basel, F Hoffmann-La Roche Ltd., Basel, Switzerland

**Abstract** Invasion of erythrocytes by *Plasmodial* merozoites is a composite process involving the interplay of several proteins. Among them, the *Plasmodium falciparum* Cysteine-Rich Protective Antigen (PfCyRPA) is a crucial component of a ternary complex, including Reticulocyte binding-like Homologous protein 5 (PfRH5) and the RH5-interacting protein (PfRipr), essential for erythrocyte invasion. Here, we present the crystal structures of PfCyRPA and its complex with the antigen-binding fragment of a parasite growth inhibitory antibody. PfCyRPA adopts a 6-bladed $\beta$-propeller structure with similarity to the classic sialidase fold, but it has no sialidase activity and fulfills a purely non-enzymatic function. Characterization of the epitope recognized by protective antibodies may facilitate design of peptidomimetics to focus vaccine responses on protective epitopes. Both in vitro and in vivo anti-PfCyRPA and anti-PfRH5 antibodies showed more potent parasite growth inhibitory activity in combination than on their own, supporting a combined delivery of PfCyRPA and PfRH5 in vaccines.

*For correspondence: Gerd. Pluschke@unibas.ch (GP); markus. rudolph@roche.com (MGR)

**Present address:** †Red de Estudios Moleculares Avanzados, Instituto de Ecología A.C, Xalapa, México

**Competing interests:** The authors declare that no competing interests exist.

## Introduction

According to the World Health Organization 2015 Malaria Report (who.int/malaria/publications/world_malaria_report/en), malaria is estimated to have caused 214 million clinical cases and 438,000 deaths in 2015. The disease is transmitted by female *Anopheles* mosquitoes and caused by parasitic protozoans of the genus *Plasmodium*, of which *P. falciparum* and *P. vivax* are the most prevalent and *P. falciparum* is causing the *most* often fatal and medically most severe form of malaria. Debilitating clinical symptoms associated with the infection are caused by the multiplication of the asexual blood-stage parasites in erythrocytes. One of the most promising targets for malaria vaccine development is therefore at the stage where merozoites invade erythrocytes.

Invasion of host erythrocytes by merozoites is a complex process, conceptually divisible into four phases: (1) initial recognition of and reversible attachment to the erythrocyte membrane by the merozoite; (2) junction formation leading to irreversible attachment of the merozoite, parasitophorous vacuole formation, and release of the *Plasmodium* rhoptry-microneme secretory organelles; (3) invagination of the erythrocyte membrane around the merozoite, accompanied by the shedding of the merozoite's surface coat; (4) closing of the parasitophorous vacuole and resealing of the erythrocyte membrane mark the completion of merozoite invasion (*Pinder et al., 2000*). The initial recognition and the active invasion of erythrocytes depend on specific molecular interactions between

**eLife digest** Malaria is one of the deadliest infectious diseases worldwide, killing over 400,000 people a year. About 200 million people are infected every year, placing a huge social and medical burden especially on developing countries. Microscopic parasites known as *Plasmodium* are responsible for causing this disease. *Plasmodium* parasites have a complex life cycle involving both mosquito and mammal hosts. This includes a stage where the parasites infect the mammal's red blood cells, which causes the symptoms of the disease. In 2012, a team of researchers discovered that a protein called CyRPA forms a group (or 'complex') with several other proteins to allow the parasites to enter red blood cells.

Developing a vaccine is one of the most promising approaches to prevent malaria. Vaccines help the body to recognise and fight an invading microbe by triggering an immune response that results in the production of proteins called antibodies, which can bind to specific molecules on the surface of the microbe. If the microbe later enters the body, these antibodies can be produced quickly to eliminate the microbe before it causes disease. However, efforts to develop a highly effective vaccine against malaria have so far been unsuccessful.

Favuzza et al. – including some of the researchers involved in the 2012 work – used a technique called X-ray crystallography to investigate the three-dimensional structure of the CyRPA protein. The experiments show that an antibody is able to bind to a region of CyRPA – a designated 'protective epitope' – that is similar in the CyRPA proteins of all *Plasmodium falciparum* strains. These antibodies can prevent the parasite from entering the red blood cells, and vaccines containing CyRPA may therefore be effective at protecting individuals from malaria. The findings of Favuzza et al. also suggest that using CyRPA in combination with another protein in the complex called RH5 could make the vaccine more powerful as it would make it harder for the parasite to become resistant.

The next step following on from this work is to design a vaccine containing protective CyRPA epitopes that triggers an immune response in mammals that is strong enough to reduce the numbers of parasites in the blood. A future challenge will be to develop a vaccine that combines several proteins involved in different stages of the parasite's life cycle to provide full protection against malaria.

parasite ligands and receptors on the host erythrocyte membrane. Although several ligand-receptor interactions have already been identified, the entire network of molecular interactions involved in invasion is not yet fully disentangled. In addition, *P. falciparum* merozoite proteins are antigenically highly diverse and in part functionally redundant, to facilitate parasite escape from host immune surveillance and to ensure erythrocyte invasion via alternative pathways (*Cowman et al., 2012*).

Most efforts in malaria blood stage vaccine research and development have historically concentrated on immuno-dominant, polymorphic antigens that contribute to the invasion of red blood cells by merozoites. Despite major efforts, blood-stage vaccines based on merozoite surface antigens have so far shown limited efficacy in clinical trials (reviewed in *Halbroth and Draper, 2015*). Extensive antigenic polymorphism represents one major hurdle for the development of an effective blood-stage malaria vaccine (*Takala et al., 2009*; *Dzikowski and Deitsch, 2009*). Therefore, the identification of new candidate antigens that are able to induce broad strain-transcending immunity and that are not susceptible to 'vaccine resistance' has become a recent research focus.

Availability of pathogen genomes is facilitating the discovery of novel vaccine candidate antigens through 'reverse vaccinology' approaches (*Rappuoli, 2001*; *Donati and Rappuoli, 2013*). Sequencing and annotation of the *P. falciparum* genome (*Gardner et al., 2002*) has supported the identification of new blood-stage vaccine candidate antigens (*Conway, 2015*; *Proietti and Doolan, 2014*), among which the *P. falciparum* Cysteine-Rich Protective Antigen (PfCyRPA) has a number of noteworthy properties. While PfCyRPA is highly conserved among a plethora of *P. falciparum* isolates, it also is poorly immunogenic in the context of natural exposure (*Dreyer et al., 2012*). Moreover, PfCyRPA-specific monoclonal antibodies (mAb) inhibit parasite growth both in vitro and in vivo by blocking merozoite invasion (*Dreyer et al., 2012*; *Favuzza et al., 2016*).

PfCyRPA is a 42.8 kDa protein of 362 residues with a predicted N-terminal secretion signal. Orthologs of PfCyRPA have been found in the genomes of the human malaria parasite *P. vivax* and the primate pathogens *P. knowlesi*, *P. cynomolgi*, and *P. reichenowi* (*Figure 3—figure supplement 1*), but not in the sequenced genomes of other *Plasmodium* species. *P. falciparum* PfCyRPA shares on average 42% sequence identity with its orthologs, but within different *P. falciparum* isolates PfCyRPA is highly conserved: just 13 dimorphic amino acid positions (highlighted in *Figure 3—figure supplement 1*) were found in 227 *P. falciparum* field isolates (*Manske et al., 2012*), and only a single variant (Arg399 instead of Ser399) was found at a frequency of greater than 2%.

PfCyRPA is part of a multi-protein complex (*Reddy et al., 2015*; *Volz et al., 2016*) including also the PfRH5-interacting protein PfRipr and the reticulocyte binding-like homologous protein PfRH5, which binds to the erythrocyte receptor basigin (*Baum et al., 2009*; *Crosnier et al., 2011*; *Chen et al., 2011b*, *2014*). PfRH5, PfCyRPA, and PfRipr colocalize during parasite invasion at the junction between merozoites and erythrocytes. The complex seems to be required both for triggering $Ca^{2+}$ release and establishment of tight junctions (*Volz et al., 2016*). While merozoites deficient in PfCyRPA or PfRH5 can still bind to erythrocytes, they do not attach irreversibly and cannot invade the host cells (*Volz et al., 2016*). Like PfCyRPA (*Dreyer et al., 2012*), PfRH5 induces invasion-blocking antibodies that are effective across common genetic variants (*Douglas et al., 2011*; *Bustamante et al., 2013*; *Douglas et al., 2014*).

'Structural vaccinology', a combination of immunological, structural, and bioinformatics approaches, is increasingly used for the design of improved vaccine antigens (*Dormitzer et al., 2008*; *Cozzi et al., 2013*; *Malito et al., 2015*). To this end, the crystal structures of PfRH5 in complex with basigin and neutralizing inhibitory mAb have been determined (*Chen et al., 2014*; *Wright et al., 2014*). Here, we describe the crystal structure of the promising vaccine candidate PfCyRPA alone and in complex with the antigen-binding fragment (Fab) of the parasite growth inhibitory mAb c12 (*Dreyer et al., 2012*). The structure of PfCyRPA represents a step toward elucidating its biological function. Furthermore, definition of the specific epitope–paratope interactions from the crystal structure of the PfCyRPA/c12 complex will support rational design of an epitope-focused PfCyRPA-based candidate vaccine.

## Results

### Fine specificities of anti-PfCyRPA antibodies

The finding that PfCyRPA and PfRH5 form a complex essential for parasite invasion prompted us to investigate the fine specificities of previously generated parasite inhibitory and non-inhibitory anti-PfCyRPA mAbs. The 16 anti-PfCyRPA mAbs available for analysis (*Dreyer et al., 2012*; *Favuzza et al., 2016*), showed six distinctive reactivity patterns with seven overlapping recombinant protein fragments of PfCyRPA, assigning them to the epitope groups A – F, with groups A, B, C, and F comprising the parasite inhibitory and groups D and E the non-inhibitory mAbs (*Figure 1*). All mAbs bound to the full-length PfCyRPA (without the signal sequence; fragment 1). mAbs belonging to epitope group A exclusively bound this fragment, indicating that they recognize conformational epitopes not present in any of the shorter PfCyRPA sequence stretches. Lack of binding to fragments 2 and 7 indicates that the epitope may comprise sequences from both ends of the polypeptide chains. Epitope group B antibodies, including mAb c12, bound only to fragments 1, 2, and 3. Epitope group F mAbs bound to fragments 1, 2, and 5, but not to fragment 3, indicating that in contrast to group B, residues located in the sequence stretch between aa 181–251 are required for their binding. The single mAb c04 constitutes the epitope group C showing binding to fragment 7 (only in IFA, not confirmed by Western blotting analysis). The non-inhibitory mAbs clustered into the distinct epitope groups D and E.

### The parasite growth inhibitory activity of anti-PfCyRPA antibodies is enhanced by anti- PfRH5 antibodies

Since it may prove useful to incorporate a combination of PfCyRPA and PfRH5 in a multivalent malaria vaccine, we investigated whether mAbs against these two vaccine candidate antigens have additive or synergistic effects. In a first step, we tested a combination of inhibitory mAbs against the two antigens in an in vitro parasite growth inhibition assay. Parasites were cultured for one cycle of

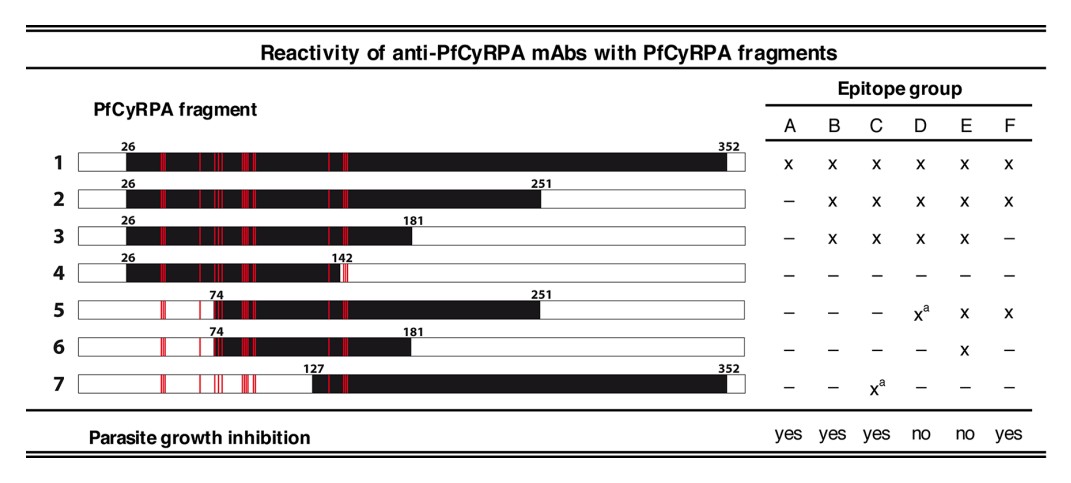

**Figure 1.** Binding of anti-PfCyRPA mAbs to fragments of PfCyRPA. Binding of 16 mAbs to PfCyRPA fragments (black bars) expressed on the cell surface of HEK cells as assessed by Western blotting analysis and live-cell immunofluorescence staining. (x) indicates staining and (–) no staining; (a) indicates no reactivity in Western blot analysis of HEK cell lysates. Expression on the surface of the HEK cells has been demonstrated for all PfCyRPA fragments by immunofluorescence analysis using anti-Histidine tag HIS-6/9 mAb (*Figure 1—figure supplement 1*). For reference, the 17 residues constituting the epitope on PfCyRPA identified from the complex crystal structure with the Fab of mAb c12 is shown in all constructs as red bars. According to their reactivity pattern, anti-PfCyRPA mAbs were assigned to different epitope groups: A: c10, SB2.5; B: c02, c06, c08, c09, c12, SB3.7; C: c04; D: c05; E: c13, SB3.9; F: SB1.6, SB2.1, SB2.3, SB3.3.

The following figure supplement is available for figure 1:

**Figure supplement 1.** Cell-surface expression of PfCyRPA fragments on transiently transfected HEK cells.

merozoite invasion in the presence of the anti-PfCyRPA c12 mAb with or without the anti-PfRH5 BS1.2 mAb at concentrations of 500, 250, and 125 µg/mL. Either mAbs showed potent inhibitory activity, consistently reducing parasite growth of all four tested *P. falciparum* strains in a concentration-dependent manner and to the same extent as the well characterized inhibitory anti-MSP-1 mAb 12.10 (*Blackman et al., 1990*) (*Figure 2A* and *Figure 2—figure supplement 1*). When combining the anti-PfCyRPA c12 mAb with the anti-RH5 BS1.2 mAb, we found a significantly enhanced inhibitory activity: while mAbs c12 and BS1.2 at a concentration of 250 µg/mL inhibited growth by 21% ± 2.2% and 31% ± 4.6%, respectively, the combination of both mAbs (250 µg/mL each) inhibited growth by 59% ± 1.4%; (*Figure 2A*). The functional activity of both mAbs was not enhanced by the addition of a malaria-unrelated control mAb.

In a second step, the in vivo parasite inhibitory activity of the mAbs was evaluated in the *P. falciparum* SCID murine model that employs non-myelodepleted NODscidIL2Rγnull mice engrafted with human erythrocytes (*Dreyer et al., 2012*; *Jiménez-Díaz et al., 2009*). Groups of three mice received 2.5 or 0.5 mg of mAbs c12 or BS1.2 or a combination of both mAbs by i.v. injection. The control groups received either 2.5 mg of an isotype-matched malaria-unrelated mAb or the same volume of PBS without Ab. Mice were infected with parasitized erythrocytes 1 day after the antibody transfer and parasitemia was subsequently monitored (*Figure 2B*). In the control groups, parasitemia reached 19.6% ± 0.8% on day 9 after mAb injection. Parasitemia in mice having received 2.5 mg c12 or BS1.2 mAb increased only marginally, reaching 2.2 ± 0.5 and 2% ± 0.3% on day nine after mAb injection, respectively. At the lower dose of 0.5 mg c12 and BS1.2 inhibited parasite growth to 10.1 ± 2.3 and 10.8% ± 6.9% parasitemia, respectively (*Figure 2B*). In accord with the in vitro data, parasitemia decreased significantly (p=0.0356; unpaired t test, 95% confidence interval, two-tailed) and reached only 4.8% ± 1.9% on day 9 if mice received 0.5 mg of the anti-PfRH5 BS1.2 mAb in addition to 0.5 mg of the anti-PfCyRPA c12 mAb.

These results demonstrated that anti-PfCyRPA and anti-PfRH5 antibodies have an additive parasite growth inhibitory effect, justifying the combination of both antigens in a subunit vaccine. While the structure of PfRH5 in complex with an inhibitory antibody has been determined, structural

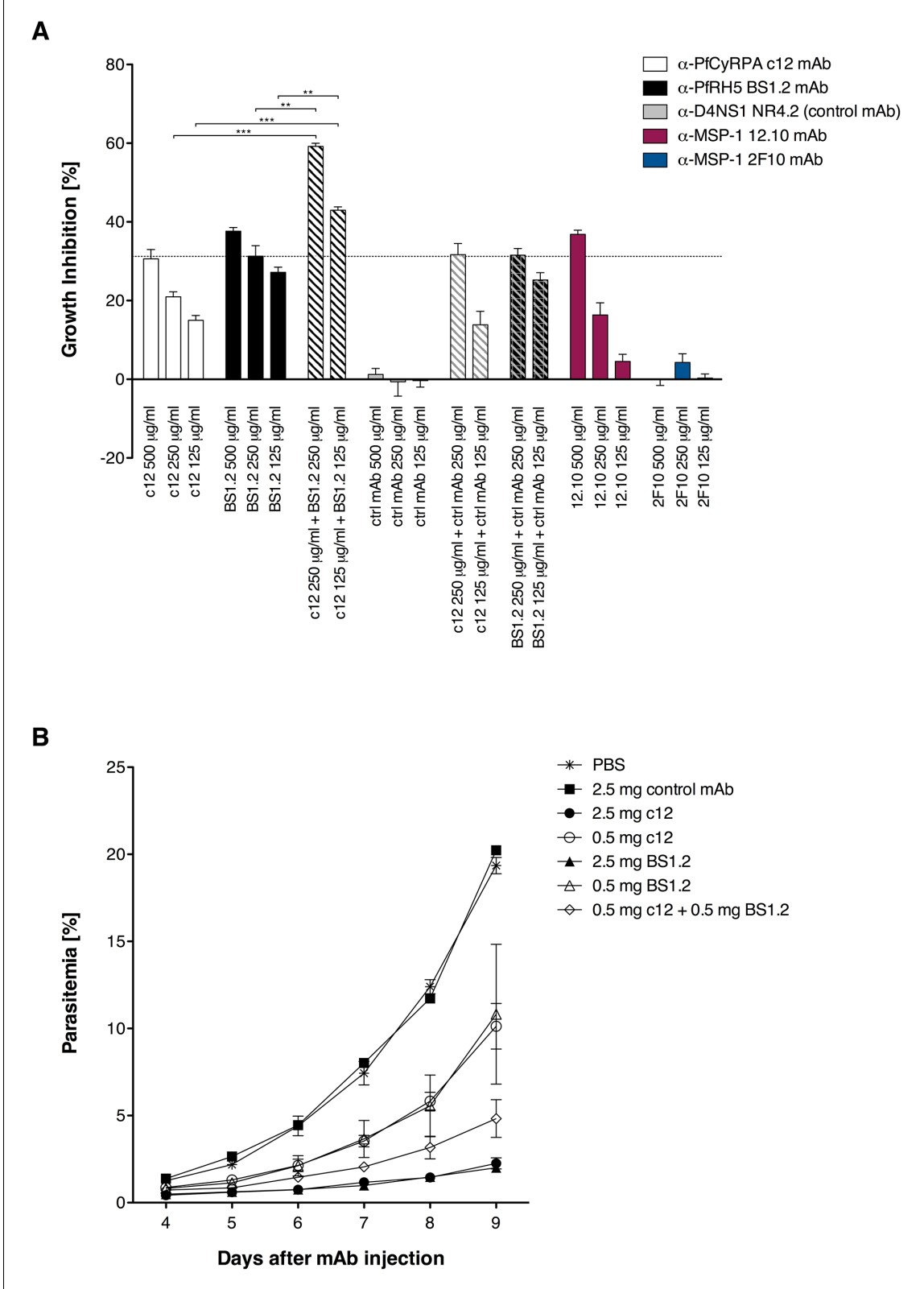

**Figure 2.** Anti-PfCyRPA and anti-PfRH5 mAbs have both in vitro and in vivo an additive inhibitory effect on parasite growth. (**A**) Growth inhibition in vitro. Synchronized *P. falciparum* 3D7 blood-stage parasites were cultured for one cycle of merozoite invasion (48 hr) in the presence of anti-PfCyRPA c12 mAb, anti-PfRH5 BS1.2 mAb, and their combinations. An isotype-matched, malaria-unrelated mAb (NR4.2) (*Rose et al., 2016*) was used as negative control. Inhibitory and non-inhibitory anti-MSP-1 mAbs (12.10 and 2F10, respectively) were also included as reference (*Blackman et al., 1990*,

*Figure 2 continued on next page*

*Figure 2 continued*

*1994*). Percent parasite growth inhibition was calculated against the parasitemia of PBS control wells. Each bar represents the mean of a triplicate experiment, and error bars indicate the standard deviation (SD). Differences in parasite growth inhibition between mAbs c12 and BS1.2 alone and their combinations are statistically significant (unpaired t test with Welch's correction, 95% confidence interval, two-tailed p value). (B) Growth inhibition in vivo. NODscidIL2Rγnull mice received purified anti-PfCyRPA c12 mAb and/or anti-PfRH5 BS1.2 mAb by i.v. injections. Mice were then infected with *P. falciparum* 3D7 and parasitemia was monitored over 6 days. Values represent the mean parasitemia in human erythrocytes in peripheral blood of three mice per group. Error bars indicate the SD. PBS and an unrelated control mAb were used as negative control.

The following figure supplement is available for figure 2:

**Figure supplement 1.** Anti-PfCyRPA and anti-PfRH5 mAbs inhibit parasite growth of various *P. falciparum* strains.

information for PfCyRPA is lacking. We therefore determined the crystal structures of PfCyRPA and its complex with the Fab of the growth inhibitory mAb c12.

## Biophysical analysis and crystal structure of PfCyRPA

The far-UV CD spectrum of PfCyRPA is consistent with an all-β structure connected by loop regions and the absence of α-helices (*Figure 3—figure supplement 2A*). Mass spectrometric analysis of proteolytic fragments of PfCyRPA revealed at least four disulfide bonds that are sequential along the sequence (*Figure 3—figure supplement 2B*). Intrinsic fluorescence showed that the two Trp residues present in PfCyRPA are buried in the native state. Also, the disulfide bonds seem to be buried, because addition of 50 mM reducing agent had no significant effect on the fluorescence of PfCyRPA (*Figure 3—figure supplement 2C*). Taken together, these data are consistent with PfCyRPA forming a compact, disulfide-stabilized molecule of predominantly β-sheet structure.

In order to crystallize PfCyRPA, we needed to pre-treat the protein with Actinase E (*Figure 4—figure supplement 1A*). The crystal structure of PfCyRPA, determined to a resolution of 2.5 Å (detailed in *Supplementary file 1*), confirmed our biophysical analyses (*Figure 3*). PfCyRPA adopts a six-bladed β-propeller structure that buries the disulfide bonds and the Trp residues. Each blade of the propeller is constructed by a four-stranded anti-parallel β-sheet (*Figure 3B*). The five disulfide bonds in PfCyRPA are located within blades 2–6 (*Figure 3A*), stabilizing each individual blade. The first blade has no disulfide bond; it is formed by β-strands from the N- and C-terminal regions of PfCyRPA, potentially enabling conformational changes in PfCyRPA by opening and closing. A domain alignment search (DALI; [*Holm and Rosenström, 2010*]) for related structures revealed that PfCyRPA adopts a heavily modified sialidase/neuraminidase fold. The closest structural relative is the catalytic domain of *Vibrio cholerae* sialidase (*Moustafa et al., 2004*) (*Figure 3C*). The two proteins have only 9% sequence identity and the structures have a large root mean square distance (rmsd) of 3.7 Å over 285 residues, clearly showing that while the overall fold is similar, the structures are very different with respect to the inclination of the blades (*Figure 3C*) and the length and conformations of the surface loops connecting the β-strands (*Figure 3D*). PfCyRPA also contains a signature sequence motif for sialidases known as an Asp-box (201-SHDKGETW-208; conserved residues are underlined) (*Roggentin et al., 1989*), which serve structural roles in the β-propellers of sialidases. However, while bacterial sialidases contain between three to five Asp-boxes, PfCyRPA contains only a single one.

Two molecules of PfCyRPA are present in the asymmetric unit and have conformational differences in several surface loops, suggesting possible flexibility of these loops in solution (*Figure 3B*). From the point of peptidomimetics that could be derived from PfCyRPA for vaccine development, these surface loops connecting the blades on the back and front of the β-propeller are natural candidates. Although some of the loops, e.g. in blade 5 (arrows in *Figure 3B*), have significant structural plasticity, it should be possible to stabilize them in a conformation suitable to raise an immune response.

## PfCyRPA has no sialidase activity

The sialidase fold of PfCyRPA and the presence of an Asp-box motif raised the question whether PfCyRPA exhibits sialidase activity. In view of the involvement of viral and microbial sialidases in the unmasking of cryptic host ligands, host cell adhesion, and invasion (*Chen et al., 2011a*; *Lewis and*

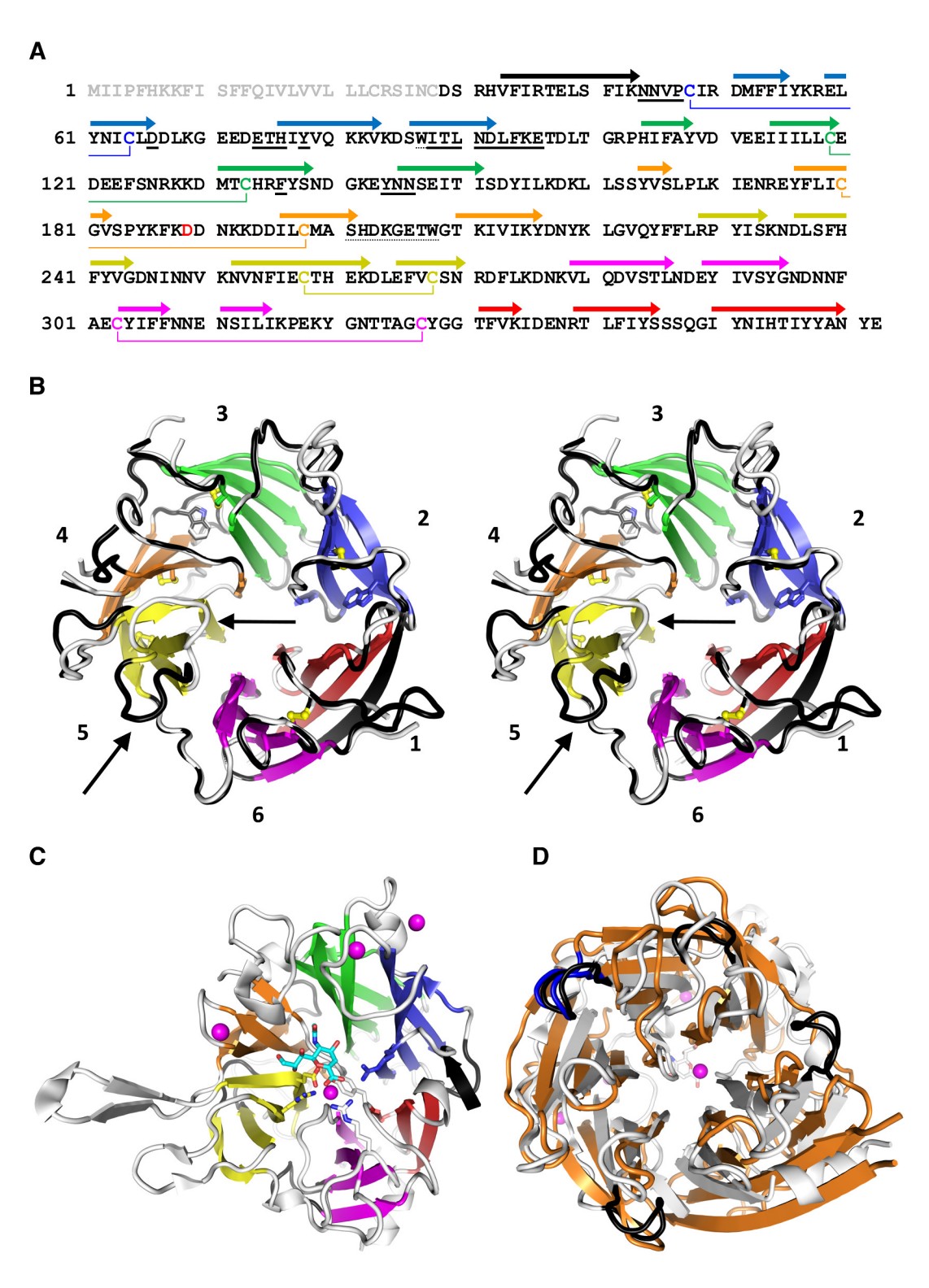

**Figure 3.** PfCyRPA adopts the neuraminidase fold. (A) Structure-sequence relationship of PfCyRPA. Indicated are an Actinase E cleavage site at Asp189 (red), a sialidase-typical Asp-box (dotted underlined), the two Trp residues (dotted underlined), and the sequential disulfide bonds (connected by lines and in same color). β-strands are shown as arrows colored according to the blade they form. The epitope recognized by mAb c12 is underlined in bold. (B) Cross-eyed stereo view of the ribbon representation of a superposition of the two PfCyRPA molecules in the asymmetric unit with the blades

*Figure 3 continued on next page*

*Figure 3 continued*

numbered 1–6 from the N-terminus and colored individually. Blade one is made up of an N-terminal (black) and three C-terminal β-strands (red). One protomer is shown with white, the other with black loop regions, which may differ substantially (arrows in blade 5). The Trp and Cys residues are drawn as stick models. (C) The same orientation of the catalytic domain of *Vibrio cholerae* sialidase (PDB-ID 1w0o), the next structural homolog of PfCyRPA with a DALI score of 18 (Z < 5 is structurally dissimilar). Sialic acid and residues in the *Vibrio* enzyme are displayed as balls and sticks. Structural $Ca^{2+}$ ions are marked as magenta spheres. None of the residues necessary for metal ion binding, substrate binding, or catalysis is present in PfCyRPA. (D) Superposition of PfCyRPA with the *V. cholerae* sialidase. While both proteins are 6-bladed β-propellers, the blades have very different angles, extents, and loop lengths and conformations connecting the β-strands. The four Asp-boxes in the bacterial sialidase (grey) are colored black. PfCyRPA (orange) has only a single Asp-box connecting the third and fourth β-strands in blade 3 (colored blue). Other β-strand connections are made by sequences unrelated to the Asp-box motif, in agreement with poor conservation of the Asp-box in other, e.g. viral, sialidases. The view in (D) is rotated by 180° about the horizontal axis compared to (B) and (C).

The following figure supplements are available for figure 3:

**Figure supplement 1.** Sequence alignment of CyRPA orthologs.

**Figure supplement 2.** Biophysical analysis of PfCyRPA.

**Figure supplement 3.** PfCyRPA lacks detectable sialidase activity in a functional assay.

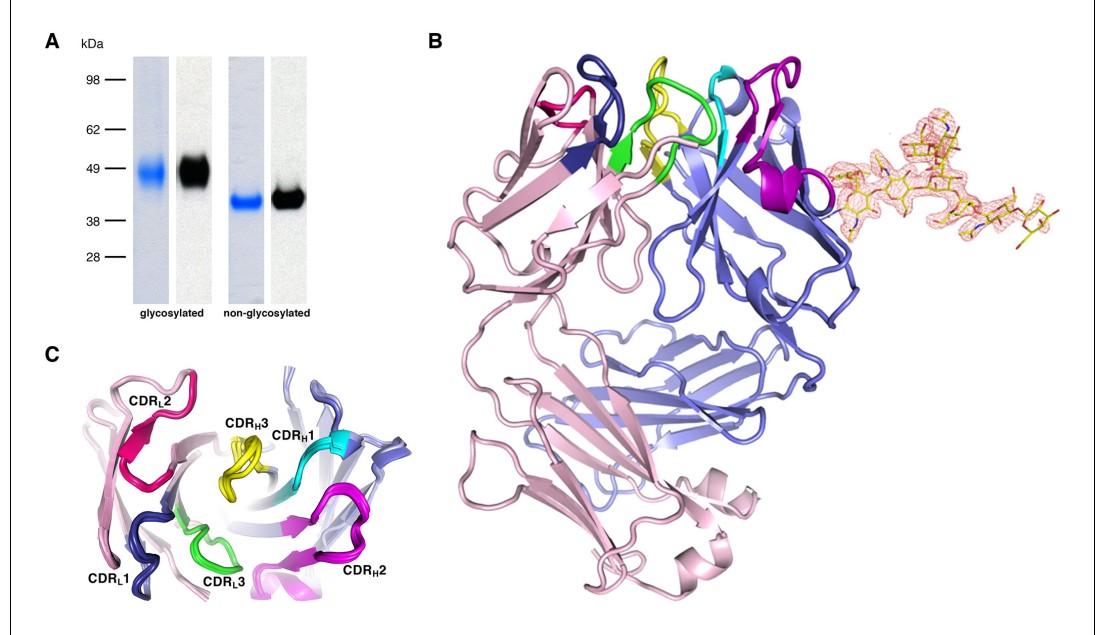

**Figure 4.** Recognition of PfCyRPA by the mAb c12 and structure of c12. (A) Reducing SDS-PAGE of glycosylated (left) and non-glycosylated (right) PfCyRPA detected by Coomassie-staining (blue) and Western blotting with mAb c12 (black). Recognition by c12 is independent of the glycosylation. (B) Overview of the c12 structure with a glycan located at heavy-chain Asn37. mFo-DFc electron density for the glycan is shown as a red mesh drawn at the three rmsd level. The light and heavy chains are colored light pink and light blue, respectively. Heavy chain CDR1-3 are colored cyan, magenta, and yellow, and light chain CDR1-3 are drawn in dark blue, pink, and green. (C) Comparison of the four c12 structures shows little conformational variability of the CDR loops. The four c12 molecules superimpose onto their variable $V_H V_L$ di-domains with an average rmsd of 0.35 Å, which reveals a minor spread of the elbow angles between 133.1° and 135.8°. The high structural congruence indicates that the CDR conformations are genuine and not dominated by crystal contacts. The view is from above on top of the CDR loops.

The following figure supplement is available for figure 4:

**Figure supplement 1.** Limited proteolysis of PfCyRPA and the PfCyRPA/c12 complex.

*Lewis, 2012*; *Matrosovich et al., 2015*), sialidase activity could make sense for PfCyRPA-mediated invasion of erythrocytes. We detected no sialidase activity, however, when we tested recombinant PfCyRPA for neuraminidase activity using a colorimetric assay (*Figure 3—figure supplement 3*). The active site of sialidases usually contains a triad of Arg residues that bind to the substrate, a Glu/Tyr pair where the acid acts as a general base to activate the Tyr nucleophile, and a hydrophobic pocket with a conserved Trp that accommodates the acetyl group of sialic acid (*Buschiazzo and Alzari, 2008*). In addition, many sialidases bind to $Ca^{2+}$ ions, of which one is close to the active site and is bound by three main-chain carbonyl groups and oxygen atoms in the side-chains of Asn, Thr, and Asp. PfCyRPA contains none of the residues necessary for catalysis, nor does it harbor a $Ca^{2+}$ binding site, providing structural correlates of the absence of sialidase activity in PfCyRPA. PfCyRPA and sialidases may have evolved from a common ancestor, or PfCyRPA could have evolved from a genuine sialidase to adopt other functionalities.

## Characterization of the epitope recognized by the parasite growth inhibitory anti-PfCyRPA mAb c12

The parasite inhibitory anti-PfCyRPA c12 mAb binds tightly to PfCyRPA with a $K_d$ of ca. 1 nM as determined by surface plasmon resonance analysis (*Dreyer et al., 2012*). This mAb recognizes PfCyRPA independent of its glycosylation, as revealed by Western blot analyses (*Figure 4A*). Because of these favorable properties, we chose c12 for epitope mapping by determining the structure of a PfCyRPA/c12 complex. In a first step, we determined the crystal structure of c12 in isolation (*Figure 4B and C*). We obtained three different crystal forms containing a total of four crystallographically independent Fab molecules. Superposition of the structures showed very little structural plasticity of the CDR loops (*Figure 4C*), suggesting that they retain their structures upon epitope binding.

We then purified the PfCyRPA/c12 complex, analyzed it by limited proteolysis, and crystallized it. Actinase E treatment of the PfCyRPA/c12 complex resulted in the same proteolysis pattern as observed for PfCyRPA alone (*Figure 4—figure supplement 1B*), suggesting that the epitope for mAb c12 is distant from the Actinase E recognition site at Asp189. Actinase E treatment was not necessary to crystallize the PfCyRPA/c12 complex, however. We determined the crystal structure of the PfCyRPA/c12 complex at a resolution of 4.0 Å by molecular replacement using the individual high-resolution structures of c12 and PfCyRPA as search models (*Figure 5A*). Despite the limited resolution, novel molecular features were visible in the electron density maps of the complex, providing confidence in the relative orientation of PfCyRPA and c12 (*Figure 5—figure supplement 1*). First, the electron density visible for PfCyRPA after molecular replacement with c12 was used as a search model for structure determination of PfCyRPA alone (see Materials and methods). This strategy would have been impossible had the placement of c12 been wrong. Second, a loop region that is absent in the PfCyRPA search model due to Actinase E proteolysis (see above) exhibits omit electron density in the PfCyRPA/c12 complex (*Figure 4—figure supplement 1*). Third, consistent with the similar conformations of the CDR loops in the individual c12 Fab structures (*Figure 4*) there are negligible conformational changes in the c12 CDR loops when bound to PfCyRPA. Similarly, the epitope conformation in PfCyRPA is very similar in the unbound and complexed form. Minor adjustments of side-chains were required during rebuilding of the complex structure. While the resolution of the PfCyRPA/c12 complex is limited and many side-chains lack clear electron density, as is typical for this resolution, knowledge of the relative orientation of PfCyRPA with respect to c12 in the complex is sufficient for designing peptidomimetics to target the immune response to the protective epitope.

The epitope recognized by mAb c12 is a surface composed of blade two and part of blade 3 of PfCyRPA (*Figure 5A*). The most frequent amino acid dimorphism of PfCyRPA at position 339 is thus located outside the epitope recognized by mAb c12, consistent with the observation that mAb c12 binds to *P. falciparum* independent of the PfCyRPA variant they express (*Dreyer et al., 2012*). mAb c12 buries a total surface area of 950 $Å^2$ on PfCyRPA with the major contributor being the light chain, which buries 520 $Å^2$, while the heavy chain buries only 430 $Å^2$. The surface complementarity coefficient Sc of the complex is 0.67, a typical value for antibody-antigen interactions (*Lawrence and Colman, 1993*); a value of one would denote perfect complementarity. The light chain also forms more potential hydrogen bonds to PfCyRPA than the heavy chain (*Figure 5B*). Seven hydrogen bonds are possible between PfCyRPA and the c12 light chain. Each side-chain of light chain CDR1 residues 50-RND-52 can form a hydrogen bond with the side-chains of PfCyRPA

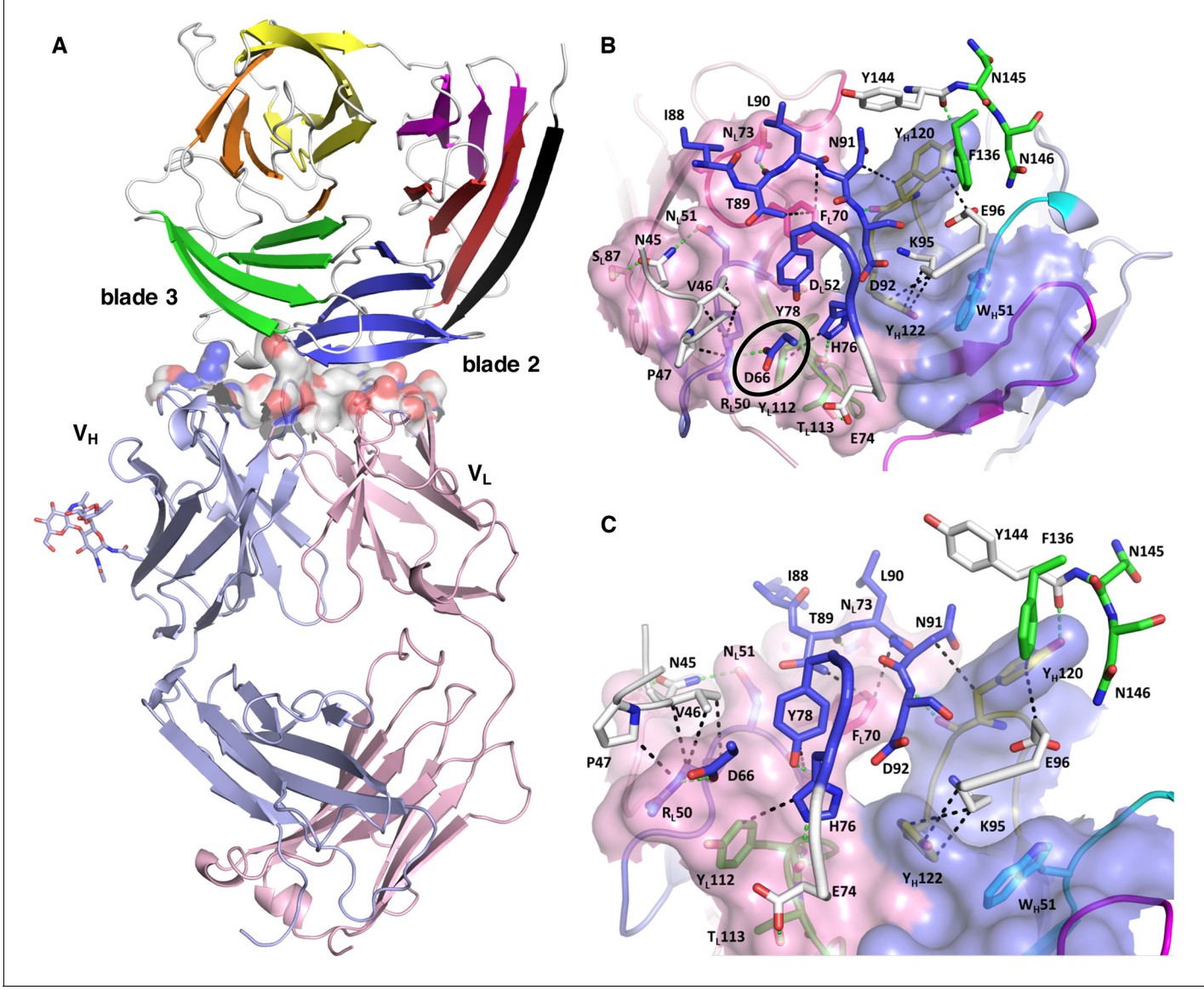

**Figure 5.** Structure of the PfCyRPA/c12 complex. (**A**) Overview showing that the majority of the interface is made by interactions between the light chain of c12 and blade 2 of PfCyRPA. (**B**) Details of the interface viewed from top onto the CDR loops. The light and heavy chain surfaces buried by PfCyRPA are colored pink and blue, respectively. Possible hydrogen bonds and van der Waals interactions are indicated by dashed green and black lines. The CDR loops are color-coded as in *Figure 4*. The Asp66-Arg50 salt bridge is circled. (**C**) Close-up of (**B**).

The following figure supplements are available for figure 5:

**Figure supplement 1.** Electron density of the PfCyRPA/c12 complex.

**Figure supplement 2.** Electron density of the PfCyRPA/c12 complex at the interface.

residues Asp66, Asn45, and Tyr78, respectively. A particularly large contribution to complex stability may stem from a salt bridge between PfCyRPA residue Asp66 and CDR1 residue Arg50, which by mutagenesis was found to be true (see below), a further confirmation that the PfCyRPA/c12 complex structure is correct. In addition, the heavy chain contributes significantly to antigen binding. Tyr120 of heavy chain CDR3 can engage in two hydrogen bonds with the side-chains of PfCyRPA residues Asp92 in blade two and Tyr144 in blade 3, thus establishing a mini-network that bridges blades 2

and 3. Together with heavy chain CDR3 residue Tyr122, Tyr120 also forms numerous van der Waals interactions with Asn91, Lys95, Glu96, and Phe136, Tyr144, Asn145, and Asn146 (*Figure 5B and C*). A complete listing of the derived potential interactions is available in *Supplementary file 2*, which due to the limited resolution of the complex structure must remain tentative. What is clear from the structure is that mAb c12 recognizes a discontinuous epitope that contains at least 17 residues distributed over four PfCyRPA sequence stretches (marked in *Figure 3A*).

The epitope extracted from the complex structure matches the binding pattern of mAb c12 to the PfCyRPA fragments in our epitope analysis (*Figure 1*): only fragments 1–3 containing all the seventeen interaction sites were recognized by mAb c12. Furthermore, fragment 4, lacking three out of these 17 residues (Tyr144, Asn145, and Asn146) and fragments 5 and 6, lacking four residues (Asn45, Val46, Pro47, and Asp66), were not recognized by mAb c12.

The PfCyRPA epitope recognized by mAb c12 was further verified by mutagenesis. The crystal structure suggests that Asp66 forms a salt bridge with Arg50 in the CDR1 loop of the c12 light chain (*Supplementary file 2* and *Figure 5—figure supplement 1*). When Asp66 was changed to Lys, a capture ELISA detected significantly weaker binding of the mAb c12 to the Asp66Lys variant of PfCyRPA (*Figure 6A*). As expected, the Asp66Lys replacement did not affect binding of the control mAb SB1.6 (*Figure 6B*), which belongs to the unrelated epitope group F and is expected to engage in crucial interactions with PfCyRPA residues located between residues 181 and 251 (*Figure 1*).

## Discussion

PfCyRPA has been identified as a novel malaria blood-stage vaccine target in an endeavor to test predicted *P. falciparum* open reading frames for the capacity to elicit parasite-inhibitory mAbs (*Dreyer et al., 2010*). PfCyRPA is stage-specifically expressed in late schizonts and elicits mAbs that inhibit parasite growth in vitro and in a *P. falciparum* experimental infection model based on NODscidIL2Rγnull mice engrafted with human erythrocytes (*Dreyer et al., 2012*). Recent studies (*Reddy et al., 2015*) have placed PfCyRPA in a ternary complex with PfRH5 and PfRipr that as a whole is essential for erythrocyte invasion (*Volz et al., 2016*). Like PfRH5 and PfRipr (*Baum et al., 2009*; *Chen et al., 2011b*), PfCyRPA is refractory to genetic disruption (*Reddy et al., 2015*), and loss of PfCyRPA or PFRH5 function in conditionally expressing mutants blocks parasite growth due

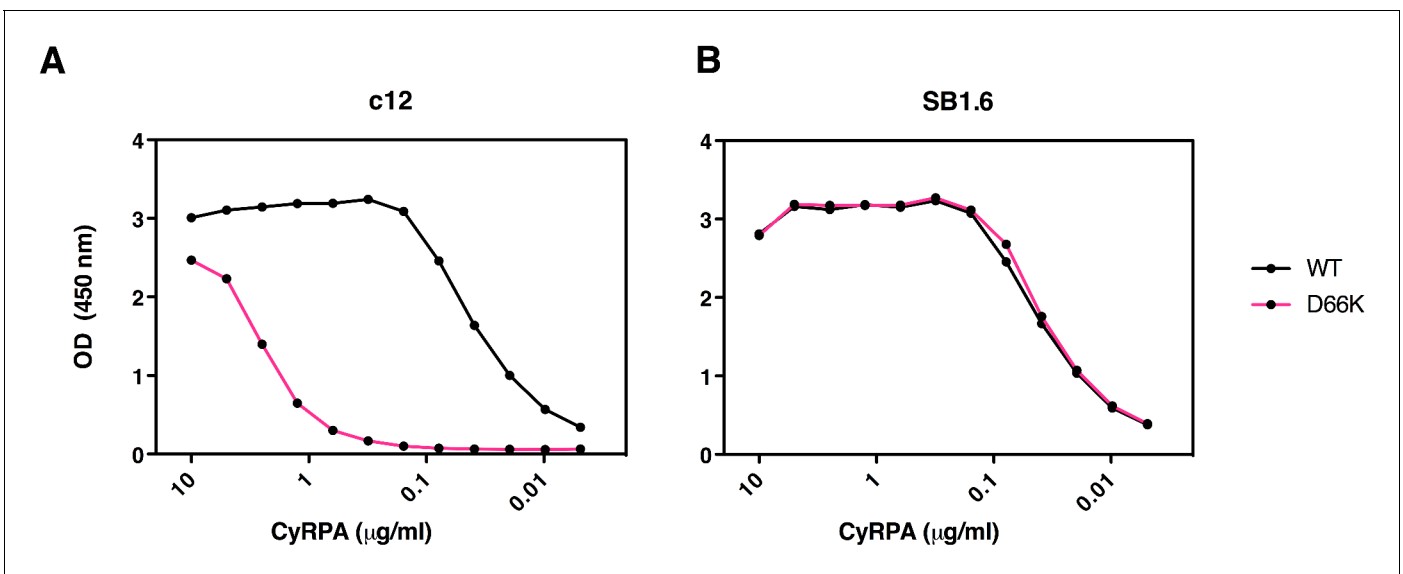

**Figure 6.** Identification of Asp66 as a key contact residue for PfCyRPA/c12 interaction. mAb binding to purified wild-type PfCyRPA and an Asp66Lys single amino acid sequence variant was analyzed by capture-ELISA experiments. ELISA plates were coated with 10 μg/mL anti-PfCyRPA mAbs and then incubated with serial dilutions of wild-type PfCyRPA (WT) or an Asp66Lys variant (D66K); HRPO-labeled anti-Histidine tag mAbs were used as detection antibody. (**A**) When compared to wild-type PfCyRPA, the Asp66Lys single amino acid exchange strongly reduced the binding to mAb c12 mAb (epitope group B; *Figure 1*). (**B**) In contrast, the Asp66Lys amino acid exchange did not affect the binding of the mAb SB1.6, which belongs to epitope group F.

to the inability of merozoites to invade erythrocytes (*Volz et al., 2016*). Here, we show that antibodies against PfRH5 and PfCyRPA have additive parasite growth inhibitory activity. Inclusion of both antigens into a multivalent subunit vaccine therefore represents an attractive strategy to prevent emergence of escape mutants. While an ortholog of PfCyRPA is also present in the genome of the human parasite *P. vivax*, an ortholog of PfRH5 has only been found so far in the genome of *P. reichenowi*, a chimpanzee malaria parasite closely related to *P. falciparum* (*Otto et al., 2014*). This may indicate a function of PfCyRPA beyond its role as a protein-binding platform in the ternary complex with PfRH5 and PfRipr. While it has been claimed that PfCyRPA anchors the ternary complex through a GPI anchor to the parasite membrane (*Reddy et al., 2015*), recent results are not consistent with this suggestion (*Volz et al., 2016*). This inconsistency suggests that additional proteins might be involved in anchoring the complex to the parasite membrane and that PfCyRPA has an essential function other than that of an anchor. Further functional studies are required to elucidate the molecular mechanisms how the ternary complex is involved in the triggering of $Ca^{2+}$ release and the establishment of tight junctions between the merozoite and the erythrocytes (*Volz et al., 2016*). In this context, it may be relevant that the five sequential disulfide bonds of PfCyRPA are in blades 1–5, leaving blade 6 of the $\beta$-propeller structure without such a local stabilization. Blade six is constructed by $\beta$-strands from the N- and the C-terminus, and in principle could act as a gate to allow PfCyRPA to undergo substantial conformational changes.

Structural vaccinology defines epitopes from antigen-antibody complex crystal structures and thus allows the design of surrogate antigens that elicit protective humoral immune responses. The approach may be of particular use for pathogens that cause chronic infections by presenting immuno-dominant antigens and epitopes to the immune system, which, however, only elicit non-protective antibody responses (*Liljeroos et al., 2015*; *Loomis and Johnson, 2015*; *Robinson, 2013*; *Dormitzer et al., 2012*). The structural studies of *Neisseria meningitidis* adhesin A and factor H binding protein (*Malito et al., 2013*), as well as of the *Staphylococcus aureus* manganese transport protein (*Ahuja et al., 2015*) represent remarkable examples in which crystal structure determination of antigens elucidated both the molecular mechanism of their biological functions and their immunological viability as vaccine antigens.

For the development of synthetic malaria subunit vaccines, conformationally stabilized and structurally optimized peptide antigens mimicking conserved protective epitopes (*Mueller et al., 2003*; *Okitsu et al., 2007*; *James et al., 2006*) may be more suitable components than entire recombinant proteins containing highly polymorphic immuno-dominant non-protective surface loops. Both PfCyRPA and PfRH5 elicit protective as well as non-protective antibodies (*Dreyer et al., 2012*; *Douglas et al., 2011*), making them candidates for structure-based approaches, that aim to focus the immune responses on protective epitopes.

Even at the comparatively moderate resolution of 4 Å, the crystal structure of the protective mAb c12 bound to PfCyRPA provides an ideal starting point for the design of a conformationally stabilized PfCyRPA-derived peptide antigen to elicit primarily protective antibodies. Particle-based antigen delivery systems such as virosomes (*Cech et al., 2011*; *Pluschke and Tamborrini, 2012*) are highly suitable to support, by repetitive epitope display, the development of strong immune responses against peptidomimetics. The epitope recognized by mAb c12 is monomorphic and, like PfCyRPA in general, little immunogenic in the natural context. Focusing the vaccine-induced immune response on this PfCyRPA epitope may confer a strong protective effect.

Malaria vaccine candidates based on single-antigens have been uniformly unsuccessful and an effective subunit vaccine will likely have to include antigens from more than one stage of the parasite life cycle (pre-erythrocytic, liver stage, and blood stage components), as well as sexual blood-stage gametocyte or mosquito-stage parasite antigens as transmission-blocking components. The *Plasmodium* genome contains more than 5000 open reading frames of which only <0.5% have been tested as vaccine candidates, so careful selection of new targets for an optimized multi-valent vaccine is called for (*Proietti and Doolan, 2014*; *Dups et al., 2014*; *Longley et al., 2015*; *Wu et al., 2015*). PfCyRPA represents one of the most promising new blood-stage vaccine candidate antigens identified with the support of –omics approaches, and there is renewed hope that an effective combination and formulation of such rationally selected antigens can lead to the development of a malaria subunit vaccine that offers high level strain-transcending protection (*Halbroth and Draper, 2015*). With the availability of structural analyses for both PfCyRPA and PfRH5 in complex with an inhibitory antibody, a structure-based approach is now possible. In view of the observed additive effect of

antibodies, inclusion of immunogens representing both antigens in a subunit vaccine would make sense, thereby reducing the danger of selection of immune escape mutants.

## Materials and methods

### Molecular biology

Plasmids were amplified in *E. coli* strain Top10 (Life Technologies) grown in LB medium under 100 µg/mL ampicillin selection. The expression vector for secretion of recombinant PfCyRPA was generated by PCR using the plasmid BVM_PFD1130W_FLAG_GP_His as template (*Dreyer et al., 2010*). The resulting expression vector pcDNA3.1_BVM_CyRPA(26-362)_His$_6$ encodes the bee-venom melittin (BVM) signal sequence to secrete PfCyRPA into the cultivation medium, and a C-terminal His$_6$-tag. Three asparagine residues (N145, N322, and N338) were predicted to be potential sites for N-glycosylation of PfCyRPA in mammalian cells. Since *Plasmodium* proteins are not glycosylated (*Dieckmann-Schuppert et al., 1992*), an expression vector coding for the PfCyRPA triple variant N145Q/N322Q/N338Q comprising residues 29–362 and containing no N-glycosylation sites was derived from the first vector by site-directed mutagenesis (GenScript). The expression vector coding for the PfCyRPA Asp66Lys single amino acid variant (D66K) was also generated by site-directed mutagenesis (GenScript) resulting in the expression plasmid pcDNA3.1_BMV_CyRPA(29–362/D66K)_His$_6$.

### Monoclonal antibodies

Generation of anti-PfCyRPA mAbs has been described elsewhere (*Dreyer et al., 2010*; *Favuzza et al., 2016*). Anti-PfRH5 antibodies were generated using the same strategy as described by *Dreyer et al. (2010)*.

### *Plasmodium falciparum* blood-stage culture

*Plasmodium falciparum* strains 3D7, K1, 7G8 and D6 were obtained from the Malaria Research and Reference Reagent Resource Center (MR4; Manassas, VA, USA) (MRA-102,–159, −154 and −285, respectively). Parasites were cultured essentially as described previously (*Matile and Pink, 1990*). The culture medium was supplemented with 0.5% AlbuMAX (Life Technologies) as a substitute for human serum (*Dorn et al., 1995*). Cultures were synchronized by sorbitol treatment (*Lambros and Vanderberg, 1979*). Erythrocytes for passages were obtained from the Swiss Red Cross (Switzerland).

### In vitro growth inhibition assay

In vitro growth inhibition assays with *P. falciparum* strains 3D7, K1, 7G8 and D6 were conducted essentially as described (*Persson et al., 2006*). Each culture (trophozoite stage parasites, 0.5% hematocrit, 2% parasitemia) was set up in triplicate in 96-well flat-bottomed culture plates. After 48 hr of incubation (one cycle of merozoite invasion), viable parasites were stained with hydroethidine and analyzed in a FACSscan flow cytometer (Becton Dickinson) using CellQuest software. A total of 50,000 cells per sample were analyzed. Percent inhibition was calculated from the mean parasitemia of triplicate test and control wells as:

$$Percent\ inhibition\ (\%) = control - test/(control/100)$$

Parasitemia of control samples was also determined by counting GIEMSA stained parasites.

### In vivo growth inhibition assay

All procedures involving living animals were performed in strict accordance with the Rules and Regulations for the Protection of Animal Rights (Tierschutzverordnung) of the Swiss Federal Food Safety and Veterinary Office. The protocol was granted ethical approval by the Veterinary Office of the county of Basel-Stadt, Switzerland (Permit Numbers: 2375 and 2303). Monoclonal antibodies were tested in the murine *P. falciparum* model essentially as described (*Dreyer et al., 2012*; *Jiménez-Díaz et al., 2009*). Human erythrocytes (hE) were administered daily (0.75 mL) by the i.v. or i.p. route. Mice received a single dose of mAbs formulation by i.v. injection. The following day, mice

were infected with $3 \times 10^7$ erythrocytes parasitized by *P. falciparum* PfNF54[0230/N3], a strain developed at GlaxoSmithKline (GSK) for growth in hE engrafted mice (*Jiménez-Díaz et al., 2009*). Parasitemia was monitored daily by flow cytometry over 6 days (days 4-9 after mAb injection).

## Protein production

FreeStyle 293 F cells (Invitrogen, R790-07), a variant of human embryonic kidney HEK cells, were cultured in suspension in serum-free medium (FreeStyle 293 Expression Medium, Thermo Fisher Scientific, Waltham, MA) at 37°C in a humidified incubator with 5% $CO_2$ in volumes of 1 L shake flasks (Corning; 120 rpm, 5 cm diameter) or 10 L wave bioreactors (Sartorius; 30 rpm, pH 7.2, 30% dissolved oxygen). Cells were diluted 1:2 with fresh culture medium and transfected at $1.2 \cdot 10^6$ cells/mL with 0.4 mg/L expression plasmid using a riDOM-based transfection system (*Québatte et al., 2014*). 72 hr post-transfection, cells were removed by filtration and the supernatant was concentrated with a 10K Pellicon three cassette (Millipore). The His$_6$-tagged recombinant proteins were purified by immobilized metal ion affinity chromatography on a HisTrap HP column (5 or 10 mL volume; GE-Healthcare) equilibrated with 50 mM HEPES/NaOH pH 7.2, 500 mM NaCl. After washing the column with the same buffer containing 20, 40, and 50 mM imidazole, the protein was eluted with a linear 50–500 mM imidazole gradient over 20 column volumes. The eluate was concentrated by ultra-filtration (Amicon Ultra-4 Ultracel 10K) and applied to a HiLoad 16/600 Superdex 200 gel permeation column (GE-Healthcare) equilibrated with 50 mM Tris/HCl pH 7.4, 150 mM NaCl. Homogeneity of PfCyRPA was assessed by reversed-phase chromatography (RP-HPLC) on a Poroshell 300 SB-C8 $1 \times 75$ mm$^2$ column using a $H_2O$ +0.01% TFA / Acetonitrile + 0.08% TFA gradient, and was confirmed by LC/MS intact mass analysis. Protein yields were 17 and 10 mg/Lr of culture for the glycosylated and non-glycosylated PfCyRPA, respectively.

Purified anti-PfCyRPA mAb c12 (*Dreyer et al., 2012*) was diluted to 0.5 mg/mL in 20 mM sodium phosphate pH 7.0 and cleaved overnight at 21°C with 0.01 mg/mL papain (Sigma-Aldrich) in a molar ratio of 1:20. The reaction was stopped with 0.001 mg/mL E64 inhibitor (Sigma-Aldrich) and the concentrated (Amicon Ultra-4 Ultracel 10K) hydrolysate was applied to a $1 \times 5$ cm$^2$ Toyopearl protein A (Tosoh Biosciences) column equilibrated in 20 mM sodium phosphate pH 7.0. The flow-through containing the Fab was concentrated and chromatographed on a $21.5 \times 60$ cm$^2$ TSKgel G3000SW column (Tosoh Bioscience) equilibrated with 20 mM bis-Tris propane/HCl pH 7.0, 200 mM NaCl. The purity of the Fab was determined by RP-HPLC as described above.

The complex of PfCyRPA with the Fab of mAb c12 (abbreviated as c12 in the following) was prepared with a 1.5-fold excess of PfCyRPA, which was incubated for 20 min at 21°C, concentrated as above and chromatographed via a Superdex 200 Increase 10/300 GL column (GE-Healthcare) equilibrated with 20 mM bis-Tris propane/HCl pH 7.0, 200 mM NaCl. Complex-containing fractions were pooled and analyzed for homogeneity by asymmetric flow field-flow fractionation with static multi-angle light scattering (SEC/AF4-MALS). For the expression of the Asp66Lys PfCyRPA variant and PfRH5, FreeStyle 293 F cells were cultured in suspension in serum-free medium (FreeStyle 293 Expression Medium, Thermo Fisher Scientific) at 37°C in a humidified incubator with 5% $CO_2$ in 125 mL shake flasks. Cells were diluted 1:2 with fresh culture medium and transfected at $10^6$ cells/mL with 0.5 mg/L expression plasmid using the 293fectin transfection reagent (Thermo Fisher Scientific). 72 hr post-transfection, cells were removed by centrifugation, and the His$_6$-tagged recombinant Asp66Lys PfCyRPA variant was purified by immobilized metal ion affinity chromatography on a HisTrap HP column (1 mL volume; GE-Healthcare) equilibrated with 50 mM Na-phosphate buffer pH 7.2, 500 mM NaCl. After washing the column with the same buffer containing 20 mM imidazole, the protein was eluted (isocratic elution) with 500 mM imidazole over five column volumes.

FreeStyle 293 F cells were tested and shown to be free of mycoplasma using MycoAlert Mycoplasma detection kit (Lonza; LT07-318). Identity of cells was confirmed using STR-PCR (Qiagen; Investigator Idplex Pkus Kit, 381625) from genomic DNA purified using High Pure PCR Template Kit (Roche Applied Science 11796828001). The obtained profile was compared to the database available from DSMZ and found to match the HEK293 profile.

## Expression of PfCyRPA fragments on the surface of HEK cells

293 HEK cells (ATCC, CRL-1573) expressing PfCyRPA fragments on the cell surface were generated essentially as described previously by *Dreyer et al. (2012)*. Briefly, DNA sequences coding for the

fragments of PfCyRPA were amplified by PCR from the BVM_PFD1130W_FLAG_GP_His plasmid (*Dreyer et al., 2010*). These expression vectors allow the anchoring of the protein of interest on the cell surface via the transmembrane domain of mouse glycophorin-A (GP). In addition, they contain the secretion signal of bee-venom melittin (BMV), a FLAG tag located extracellularly, and a His$_6$-tag located in the cytosol. The 293 HEK cells were transfected with the different expression vectors using JetPEI transfection reagent (PolyPlus) according to the manufacturer's instructions. Transient transfectants were harvested 48 hr post-transfection. HEK cell lysates were prepared at $10^7$ cells/mL in RIPA-Buffer (1 % NP40, 0.25% DOC, 10% glycerol, 2 mM EDTA, 137 mM NaCl, 20 mM Tris/HCl pH 8.0, plus protease inhibitors) and used for Western blot analysis as described in *Favuzza et al. (2016)*.

293 HEK cells were regularly tested by PCR with mycoplasma-specific primer GPO-1 (5'- ACTCC TACGGGAGGCAGCAGTA-3') and MGSO (5'-TGCACCATCTGTCACTCTGTTAACCTC-3') and shown to be mycoplasma-free.

## Biophysical characterization of PfCyRPA

To locate the disulfide bonds in PfCyRPA, peptides derived from proteolysis under native conditions by the endo-proteases LysC, AspN, and trypsin were analyzed by UPLC-tandem mass spectrometry using a Dionex UltiMate3000 RRLC system (Thermo Scientific) coupled to a Synapt G2 HDMS mass spectrometer (Waters) with an electrospray ion source and working in resolution mode under default parameters. 10 µg PfCyRPA was hydrolyzed for several hours at 37°C with 0.3 µg of LysC, trypsin, or AspN. Peptides were separated on a Waters BEH130 C18 1.7 µm UPLC column (0.3 × 150 mm$^2$) by a 5–45% gradient of acetonitrile in water containing 0.1% HCOOH at a flow rate of 10 µL/min over 90 min and eluted into the mass spectrometer. Data were analyzed using BiopharmaLynx software Ver. 1.3 (Waters). The data derived from the proteolysis are summarized in *Figure 3—figure supplement 2*.

The secondary structure composition of PfCyRPA was estimated by far-UV circular dichroism (CD) spectroscopy on a Jasco J-815 spectropolarimeter using 10 µM PfCyRPA in phosphate buffered saline (PBS) at 20°C. A cell of 1 mm path length was used to record four spectra between 200 and 320 nm with a step size of 0.1 nm and an integration time of 1 s. The spectral average was corrected for buffer contributions.

Intrinsic tryptophan fluorescence of a 5 µM PfCyRPA solution under native (PBS) and denaturing (70°C or PBS with 9 M urea) conditions was measured on an ISS PC-1 photon counting spectrofluorimeter (ISS, Inc.) at 20°C. Fluorescence emission was excited at 295 nm with 4.8 nm band width and path length 1 mm. Emission was monitored in 2 nm steps between 300 and 400 nm with a band width of 16 nm, a path length 5 mm, and an integration time of and 1 s.

## Crystallization and data collection

All crystallization were done at 21°C in the sitting drop vapor diffusion setup using a Mosquito LCP crystallization robot (TTP Labtech). If not stated otherwise crystals were cryo-protected with paraffin oil and vitrified in liquid N$_2$. A non-glycosylated variant of PfCyRPA (residues Asp29–Glu362, N145Q/N322Q/N338Q) did not yield well-diffracting crystals unless pre-treated with proteases. 1 mg/mL PfCyRPA in 50 mM Tris/HCl pH 7.4, 150 mM NaCl was diluted 1:1 with 10 mM HEPES/ NaOH pH 7.5, 500 mM NaCl, subjected for 16 hr to a panel of twelve proteases (Hampton Research Proti-Ace Kit) at a final concentration of 0.01 mg/mL, and analyzed by reducing SDS-PAGE (*Figure 4—figure supplement 1*). Of the proteases that changed the apparent molecular mass of PfCyRPA, subtilisin and proteinase K produced multiple bands, while Glu-C, elastase, and thermolysin produced a single band of slightly smaller apparent molecular mass. The thermolysin-treated PfCyRPA was subjected to crystallization, yielding triclinic crystals upon mixing of 120 nL 7.3 mg/mL PfCyRPA in 50 mM Tris/HCl pH 7.4, 150 mM NaCl with 80 nL of precipitant consisting of 0.2 M LiOAc and 20% (w/v) PEG 3350. These crystals are consistent with four molecules per unit cell but did not diffract X-rays beyond a resolution of 3.3 Å. Actinase E-treatment of PfCyRPA results in a different proteolysis pattern with two bands of apparent molecular mass 17 kDa and 20 kDa (*Figure 4— figure supplement 1*). Thin, plate-shaped monoclinic crystals diffracting to a maximum resolution of about 2.5 Å were obtained from mixing of 100 or 140 nL 15 mg/mL Actinase E-treated PfCyRPA in 50 mM HEPES/NaOH pH 7.5, 150 mM NaCl with 100 nL and 60 nL precipitants consisting of 0–0.5

M MgCl$_2$ and 20–25% PEG 3350. Crystals of the same habit were also obtained from 7.6 mg/mL PfCyRPA mixed with 19% PEG 3350, 0.3 M MgCl$_2$ and additional 5 mM CaCl$_2$, SrCl$_2$, and BaCl$_2$. Data collected from crystals grown in the presence of the heavier cations were tested for anomalous and isomorphous information content for phasing, which turned out to be negative.

Fab of mAb c12 crystallized from several conditions and its structure was determined in three crystal settings. Crystals were obtained by mixing of 80 nL 16.6 mg/mL c12 in 20 mM bis-Tris propane/HCl pH 7.0, 200 mM NaCl with 120 nL of precipitants consisting of 1.6 M sodium citrate pH 6.5 (hexagonal crystal form) or 0.1 M HEPES/NaOH pH 7.0, 20% w/v PEG 8000 (monoclinic and orthorhombic crystal forms). Monoclinic crystals were cryo-protected with reservoir solution supplemented with 20% glycerol.

Needle-shaped crystals of the PfCyRPA/c12 complex were obtained by mixing 120 nL of a 15.1 mg/mL solution in 50 mM Tris/HCl pH 7.4, 150 mM NaCl with 80 nL reservoir consisting of 0.1 M Sodium citrate pH 5.5, 17 % w/v PEG 5000 MME, 0.2 M NDSB-201 (non-detergent sulfobetaine). Cryo-protection was achieved with reservoir solution supplemented with 20% ethylene glycol.

Diffraction data were collected at Swiss Light Source beamline PX-II on a Pilatus 6M single photon counting detector using 1 Å radiation over a total range of at least 180° in fine slicing mode ($\Delta\phi$ = 0.25°). Data were indexed, integrated, and scaled with XDS (*Kabsch, 2010*), except for data from the hexagonal form of c12, which was integrated with MOSFLM and scaled with AIMLESS (*The collaborative computational project number 4, 1994*). The high-resolution limit was chosen as the shell where the correlation coefficient for half datasets CC$_{1/2}$ dropped below 70%. Data sets were tested for internal symmetry by self-Patterson and self-rotation function analyses (not shown). Twinning was excluded based on L-values and second moments. Data collection and also refinement statistics are collected in S1 Table.

## Structure determination and refinement

The presence of a short sequence motif in PfCyRPA that is characteristic for sialidases suggested that a six-bladed $\beta$-propeller might serve as a molecular replacement model. However, all attempts using several hundred $\beta$-propeller structures as search models both with and without loop regions and/or as poly-Ala models were unsuccessful. Sialidases often contain structural Ca$^{2+}$ ions, which prompted co-crystallization of PfCyRPA with the earth alkali cations Ca$^{2+}$, Sr$^{2+}$, and Ba$^{2+}$ for SIRAS phasing, but none of these provided useful derivatives. Attempts to phase the PfCyRPA diffraction data by sulfur-SAD were also unsuccessful. Packing density estimations indicated two molecules of PfCyRPA in the monoclinic asymmetric unit. The PfCyRPA data could be phased by molecular replacement using the unexplained electron density of the PfCyRPA/c12 complex (see below) as a search model. The volume of density was separated using a mask, placed in a large cubic cell, and back transformed to yield structure factor amplitudes, which were then used in a PHASER (*The collaborative computational project number 4, 1994*) molecular replacement search for two molecules. The solution had a non-random log-likelihood gain of LLG = 175 and the orientations of the two entities replicated the 180° two-fold NCS (non-crystallographic symmetry) estimated from the self-rotation function. Density averaging using PARROT (*The collaborative computational project number 4, 1994*) and the NCS operator defined by the two PHASER solutions resulted in manually interpretable electron density maps. The final model has a discontinuity at Asp189, which is consistent with cleavage of a surface loop by Actinase E. A symmetry-related PfCyRPA occupies the space liberated by the cleaved loop, explaining the necessity of the proteolytic pre-treatment for crystal formation in this setting. The model was refined using automatically generated NCS restraints excluding diverging surface loop regions. 99.5% of all residues are in the favored regions of the Ramachandran plot. The hexagonal dataset of c12 (*Supplementary file 1*) was phased by molecular replacement using a homology model generated from the c12 sequence in MOE (Chemical Computing Group) as the search model. Separate searches for the variable and constant parts of the Fab were performed in PHASER, which, as anticipated, placed the V$_H$V$_L$ and C$_H$C$_L$ domain boundaries close to each other to generate a complete Fab. The final model has 99.2% of all residues in the favored regions of the Ramachandran plot. The monoclinic and orthorhombic c12 datasets were phased using the refined hexagonal c12 structure as molecular replacement search model. Both structures have 98% of their main-chain torsion angles in the favored region of the Ramachandran plot. Elbow angles were calculated with PHENIX (*Adams et al., 2010*).

According to the Matthews parameter, the 4 Å dataset collected from the PfCyRPA/c12 crystal has enough space per asymmetric unit to harbor a single complex. Molecular replacement readily placed c12, and initial maps calculated from this partial solution revealed additional electron density at the tips of the Fab (*Figure 5—figure supplement 1*), which occupied a volume large enough to host PfCyRPA. The density resembled a $\beta$-propeller but was not interpretable due to the limited resolution of the data. However, this density provided enough phasing power to solve the 2.5 Å PfCyRPA structure (see above), showing that the placement of the Fab was correct. The same arrangement of $V_HV_L$ and $C_HC_L$ was obtained when using the $V_HV_L$ and $C_HC_L$ parts of the Fab as separate search models, indicating that the elbow angle of c12 does not change upon binding to PfCyRPA. The final model of PfCyRPA was used to complete the molecular replacement phasing of the PfCyRPA/c12 complex (log-likelihood gain of >1260), density for which is shown in *Figure 5— figure supplement 2*. A loop region not present in the PfCyRPA model could be traced in the electron density of the complex (Panel B in *Figure 5—figure supplement 1*), establishing confidence in the correctness of the molecular replacement solution. The model of the complex was refined according to established protocols for lower resolution structures (*DeLaBarre and Brunger, 2006*), that is using simulated annealing, group ADP values, automatically assigned TLS definitions, secondary structure restraints, and external restraints provided by the higher resolution individual structures of PfCyRPA and c12. No real-space refinement was applied during building and refinement of the PfCyRPA/c12 complex. Strong geometric restraints were applied throughout model building of the PfCyRPA/c12 complex. All models were built with COOT (*Emsley et al., 2010*) and refined with PHENIX (*Adams et al., 2010*).

## Acknowledgements

We thank Marcello Foggetta and Martin Siegrist for cell transfection. We also thank the staff at the Swiss Light Source for beamline support and Expose GmbH for data collection.

## Additional information

### Funding

| Funder | Author |
| --- | --- |
| Uniscientia Stiftung | Gerd Pluschke |

The funders had no role in study design, data collection and interpretation, or the decision to submit the work for publication.

### Author ORCIDs

Paola Favuzza, http://orcid.org/0000-0002-1394-927X
Gerd Pluschke, http://orcid.org/0000-0003-1957-2925

### Ethics

Animal experimentation: All procedures involving living animals were performed in strict accordance with the Rules and Regulations for the Protection of Animal Rights (Tierschutzverordnung) of the Swiss Federal Food Safety and Veterinary Office. The protocol was granted ethical approval by the Veterinary Office of the county of Basel-Stadt, Switzerland (Permit Numbers: 2375 and 2303).

### Author contributions

PF, MGR, Conception and design, Acquisition of data, Analysis and interpretation of data, Drafting or revising the article; EG, MT, BS, ACR, JE, JH, GS, BG, CJ, Acquisition of data, Analysis and interpretation of data, Drafting or revising the article; AMD, RT, JB, HM, GP, Conception and design, Analysis and interpretation of data, Drafting or revising the article; AL, Analysis and interpretation of data, Drafting or revising the article

## Additional files

### Supplementary files

• Supplementary file 1. Data collection and refinement statistics.

• Supplementary file 2. Interactions between PfCyRPA and the c12 Fab. Predicted interactions between PfCyRPA and the CDRs of mAb c12. The amino acid residues involved and the nature of the interactions are listed.

### Major datasets

The following datasets were generated:

| Author(s) | Year | Dataset title | Dataset URL | Database, license, and accessibility information |
|---|---|---|---|---|
| Favuzza P, Pluschke G, Rudolph MG | 2017 | Crystal Structure of PfCyRPA in complex with Fab c12 | http://www.rcsb.org/pdb/explore/explore.do?structureId=5EZO | Publicly available at the RCSB Protein Data Bank (accession no: 5EZO) |
| Favuzza P, Pluschke G, Rudolph MG | 2017 | Crystal Structure of Fab c12 | http://www.rcsb.org/pdb/explore/explore.do?structureId=5EZI | Publicly available at the RCSB Protein Data Bank (accession no: 5EZI) |
| Favuzza P, Pluschke G, Rudolph MG | 2017 | Crystal Structure of Fab c12 | http://www.rcsb.org/pdb/explore/explore.do?structureId=5EZL | Publicly available at the RCSB Protein Data Bank (accession no: 5EZL) |
| Favuzza P, Pluschke G, Rudolph MG | 2017 | Crystal Structure of Fab c12 | http://www.rcsb.org/pdb/explore/explore.do?structureId=5EZJ | Publicly available at the RCSB Protein Data Bank (accession no: 5EZJ) |
| Favuzza P, Pluschke G, Rudolph MG | 2017 | Crystal Structure of PfCyRPA | http://www.rcsb.org/pdb/explore/explore.do?structureId=5EZN | Publicly available at the RCSB Protein Data Bank (accession no: 5EZN) |

The following previously published dataset was used:

| Author(s) | Year | Dataset title | Dataset URL | Database, license, and accessibility information |
|---|---|---|---|---|
| Aurrecoechea C, Brestelli J, Brunk BP, Dommer J, Fischer S, Gajria B, Gao X, Gingle A, Grant G, Harb OS, Heiges M, Innamorato F, Iodice J, Kissinger JC, Kraemer E, Li W, Miller JA, Nayak V, Pennington C, Pinney DF, Roos DS, Ross C, Stoeckert CJ, Treatman C, Wang H | 2009 | PlasmoDB: a functional genomic database for malaria parasites | http://plasmodb.org/plasmo/app/record/gene/PF3D7_0423800 | The data at this site is provided freely for public use |

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
