## [Decision Letter]

Thank you for submitting your article "Structure of the malaria vaccine candidate antigen CyRPA and its complex with a parasite invasion inhibitory antibody" for consideration by *eLife*. Your article has been favorably evaluated by Wenhui Li (Senior Editor) and four reviewers, one of whom, Stephen C. Harrison (Reviewer #4), is a member of our Board of Reviewing Editors. The following individual involved in review of your submission has agreed to reveal their identity: Matthew K Higgins (Reviewer #2).

The reviewers have discussed the reviews with one another and the Reviewing Editor has drafted this decision to help you consider preparing a revised submission.

Summary:

CyRPA is an important red cell invasion protein that forms a complex with RH5 and RIPR. *P. falciparum* parasites use this complex to engage the host cell receptor, basigin, during parasite invasion of red cells. CyRPA and RH5 are both candidate vaccine immunogens, as antibodies directed against these proteins potently neutralize parasites. This manuscript extends our knowledge of antibody neutralization by showing that single monoclonals against CyRPA and RH5 act additively to provide enhanced neutralization. This result suggests that combined use of both proteins may be beneficial for vaccine development.

The authors report the structure of CyRPA at 2.5 Å resolution, which shows that it has a neuraminidase fold but lacks sialidase activity, thus indicating a non-enzymatic function. The authors also describe a structure of CyRPA bound with the Fab of neutralizing antibody c12, at to a reported 3.6 Å resolution, and validate the interaction site using a single point mutant and truncation analysis. For the most part the antibody data and the structure of CyRPA alone appear to be well done, but the reviewers have substantive concerns with the structure of CyRPA in complex with c12. Both structures are timely and relevant for the future of malaria vaccines, and the manuscript would have been suitable for publication in *eLife* were the structural results for the c12 complex more solid. If the structure with c12 has problems, the vaccine development focus of the manuscript is not justified and the manuscript less compelling overall. There are also concerns about the experiments on additivity of antibodies directed against RH5 and CyRPA, the other novel claim. We invite the authors to consider responding with a revised manuscript (and appropriate experiments and computations) to the concerns summarized below. We caution that relatively extensive new information will be required for a favorable outcome.

Essential revisions:

1) The expression of fragments of a single domain, even if it turns out to be one that is sequential in its fold (a β-propeller), generally gives unfolded polypeptide chain. Unless the fragments in Figure 1 have the correct S-S bonds within the blades that are present (are the chosen limits at positions between blades?), then they can only define linear epitopes – likely to be the least interesting ones. Moreover, in the cases in which there is no binding, have the authors demonstrated that the fragment is even expressed on the cell surface?

2) In Figure 1 the GIA data are shown simply as 'yes' or 'no'. The GIA assays used throughout seem to diverge from the standardised GIA assay in the NIH GIA reference laboratory used by many studies. This divergence makes it hard to compare how inhibitory the mAbs are when compared with other mAbs from published studies. The selling point for this paper is the ability to use this information for 'structural vaccinology' but without a real quantitative comparison of the mAbs used with the best from other studies, it is hard to know whether this c12 mAb is worth recapitulating in a vaccination attempt. More standardised and quantitative GIA would be important to make this conclusion. Indeed, the data in Figure 2 do suggest that rather high concentrations of these mAbs are required to see an inhibitory effect.

3) Because the CyRPA is a conserved protein across several Plasmodium species as well as *P. falciparum* isolates, at least one reviewer would have liked to see data showing whether the inhibitory mAb inhibits strains other than 3D7. There is no mention of the specificity or the istotype of the control mAbs. Finally, it is important to show that mAbs specific for proteins other than RH5 do not have the same additive effect with anti-PfCyRPA. That control is missing here.

4) A key point for vaccine development is to show sterilizing immunity by targeting RH5 and CyRPA. In Figure 3, however, the authors do not show the sterilizing immunity that would be expected if the 2.5 mg CS1.2 and 2.5 mg c12 combination was tested. Was this performed? If not, why not? Since the authors have shown previously that c12 inhibits invasion, the only novelty here is the combination.

5) The authors seem to have little experience with relatively low resolution structures, and their description of the refinement of the complex between Fab and PfCyRPA as "unsatisfactory" (subsection “Characterization of the epitope recognized by the parasite growth inhibitory anti-PfCyRPA mAb c12”, second paragraph) undermines confidence in the rest of the analysis, as one wonders whether the crystallographers involved (who are they?) really know how to look at a lower resolution map and how to avoid the pitfalls. Since both structures have been refined at quite high resolution, and since even the CDR loops of the Fabs do not vary much from crystal form to crystal form, it seems quite odd that the complex is so poorly determined. Specific points to consider are some red flags from data reduction statistics. Using the CC1/2 as the resolution cutoff criterion gives Rmeas of 59.5 in the outer bin (high for a low symmetry space group like P21) and I/sig(I) of 0.4 (and only 2.6 for the entire dataset). Since the elbow varies a bit among Fab c12 crystal forms, did the authors use the variable and constant modules as separate MR search objects when determining the structure of the complex?

Summary of required revisions:

1) Response to point (1), above, either with new data or – if a better structure of the c12:PfCyRPA complex can be achieved – by eliminating those experiments altogether, as a fully convincing structure will make them redundant.

2) Explicit defense of the variant GIA assay, or use of a more standard, quantitative assay.

3) Invasion inhibition assay with at least one more strain and inclusion, for both strains, of the stipulated control.

4) What combination of concentrations will give sterilizing immunity?

5) A better structure of the complex. Careful re-refinement of the structure seems in order, including careful re-examination of the initial MR solution. It might be better to collect more data, as redundancy, even at the current resolution, will result in more accurately determined intensities. Please provide some supplementary figures with maps and models, as was once standard, to restore confidence that the structure is correct. Show us the original MR-phased map in the region of the interface, together with the final model inside it, and then also the final 2Fo-Fc map, likewise with the model. One could still have problems even if these maps looked clear and well fit, but they will help.

---

## [Author Response]

*Essential revisions:*

*1) The expression of fragments of a single domain, even if it turns out to be one that is sequential in its fold (a β-propeller), generally gives unfolded polypeptide chain. Unless the fragments in Figure 1 have the correct S-S bonds within the blades that are present (are the chosen limits at positions between blades?), then they can only define linear epitopes – likely to be the least interesting ones.*

Information about the aa. sequence of the selected PfCyRPA fragments combined with the determined 3D structure of full-length PfCyRPA (data reported in Figure 1 and Figure 3) indicates that the tested fragments comprise the cysteine residues for correct S-S bonds stabilizing at least one, and in most cases more than one, intact blade structures. Furthermore, the HEK cell surface expression system used is highly suitable to generate correctly folded complete and truncated recombinant proteins.

In brief:

Fr.1 (26-352): corresponds to full length PfCyRPA. Blades 2, 3, 4, 5 and 6 should be perfectly folded. HEK cells expressing this fragment on their surface were used for mouse immunization: anti-PfCyRPA c12 mAb was generated in this way (Dreyer et al. 2010 and 2012 – doi: 10.1186/1472-6750-10-87 and 10.4049/jimmunol.1103177).

Fr.2 (26-251): comprises entire blades 2, 3 and 4;

Fr.3 (26-181): comprises entire blades 2 and 3;

Fr.4 (26-142): comprises entire blade 2 – in addition, the blade 3 partial sequence contains the two cysteine residues, but lacks the terminal β-strand;

Fr.5 (74-251): comprises entire blades 3 and 4;

Fr.6 (74-181): comprises entire blade 3;

Fr.7 (127-352): comprises entire blades 4, 5 and 6.

*Moreover, in the cases in which there is no binding, have the authors demonstrated that the fragment is even expressed on the cell surface?*

To demonstrate that the fragments are expressed on the surface of transfected HEK cells, we performed immuno-fluorescence staining of HEK cells with anti-His tag (HIS-6/9 mAb) and anti-PfCyRPA (c12 mAb) antibodies. With the anti-His tag antibody (A) all transfectants were stained, confirming the expression of PfCyRPA fragments on the cell surface. In contrast, the anti-PfCyRPA c12 antibody (B) only stained, as expected, transfectants expressing fragments 1, 2 and 3.

We have now included these data in the manuscript as “Figure 1—figure supplement 1”.

*2) In Figure 1 the GIA data are shown simply as 'yes' or 'no'.*

The anti-PfCyRPA mAbs were classified as inhibitory if they showed an in vitro parasite growth inhibitory activity (compared to the negative control) higher than 30% at the concentration of 500 μg/ml. Detailed comparative inhibition data for the 16 anti-PfCyRPA mAbs employed in this study have been published (Dreyer et al. 2012 – doi:10.4049/jimmunol.1103177, Favuzza et al. 2016 – doi:10.1186/s12936-016-1213-x).

*The GIA assays used throughout seem to diverge from the standardised GIA assay in the NIH GIA reference laboratory used by many studies.*

While we used so far a two cycle GIA, we have now repeated our experiments with the one cycle GIA protocol used by most research groups in the fields, and have replaced in the manuscript the two cycles data by one cycle data.

Results of the one cycle GIA diverged not strongly from the principal findings with the two cycles GIA (see Figure 7):

Author response image 1.Anti-PfCyRPA and anti-PfRH5 mAbs have in vitro additive inhibitory effect on parasite growth.Parasites were cultured for one cycle (48h) or two cycles (96h) of merozoite invasion in the presence of the anti-PfCyRPA c12 mAb with or without the anti-PfRH5 BS1.2 mAb at concentrations of 500, 250, and 125 μg/ml. Either mAb showed potent inhibitory activity, reducing parasite growth by ≥30% when tested at a concentration of 500 μg/ml. Inhibitory anti-MSP1 12.10 mAb and non-inhibitory anti-MSP-1 2F10 mAb were included in the assay as reference mAbs. When combining the anti-PfCyRPA c12 mAb with the anti-RH5 BS1.2 mAb, a significantly enhanced inhibitory activity was observed. The functional activity of both mAbs was not enhanced by the addition of the malaria-unrelated control mAb NR4.2. Differences in parasite growth inhibition between mAbs c12 and BS1.2 alone and their combinations are statistically significant (unpaired t test with Welch’s correction, 95% confidence interval, two-tailed p value); differences between mAbs c12 and BS1.2 alone and their combinations with the control mAb NR4.2 are not statistically significant.**DOI:**
http://dx.doi.org/10.7554/eLife.20383.019

We are using flow cytometry as readout, since this is used by most laboratories in the field. The assay which is now used for the described results is thus adapted to the assays used in many relevant publications including the publication of Chen et al. published in 2014 in *eLife* (doi: 10.7554/*eLife*.04187). In this paper the crystal structure of PfRH5 is described and the protein is characterized as a *P. falciparum* ligand essential for invasion of human erythrocytes. There, growth inhibitory activity of pfRH5-specific antibodies were calculated from a 1-cycle flow cytometry-based assay.

Other papers in the field using such a one cycle/flow cytometry GIA include:

Crosnier et al. 2011 (doi: 10.1038/nature10606)

Bustamante et al. 2013 (doi: 10.1016/j.vaccine.2012.10.106)

Theron et al. 2010 (doi: 10.1002/cyto.a.20972)

Douglas et al. 2014 (doi: 10.4049/jimmunol.1302045)

Baum et al. 2009 (doi:10.1016/j.ijpara.2008.10.006)

Chen et al. 2011 (doi:10.1371/journal.ppat.1002199)

The Methods section has been updated accordingly:

“in vitro growth inhibition assays with *P. falciparum* strains 3D7, K1, 7G8 and D6 were conducted essentially as described (Persson et al., 2006). […] Percent inhibition was calculated from the mean parasitemia of triplicate test and control wells as:

Percent inhibition (%) = control – test / (control / 100)

Parasitemia of control samples was also determined by counting GIEMSA stained parasites.

*This divergence makes it hard to compare how inhibitory the mAbs are when compared with other mAbs from published studies.*

Our one cycle data can now be easily compared to the published data using the same protocol. In addition, we included in our assays the classical inhibitory anti-MSP-1 12.10 mAb (Blackman MJ et al. J Exp Med. 1990 Jul 1; 172(1): 379–382) and non-inhibitory anti-MSP-1 2F10 mAb (Blackman MJ et al. J Exp Med. 1994 Jul 1;180(1):389-93.) as additional positive and negative controls, respectively.

*The selling point for this paper is the ability to use this information for 'structural vaccinology' but without a real quantitative comparison of the mAbs used with the best from other studies, it is hard to know whether this c12 mAb is worth recapitulating in a vaccination attempt. More standardised and quantitative GIA would be important to make this conclusion.*

Results can now be compared with many publications in the field (see above). Based on the inhibition data many research groups appear to work currently on RH5 and CyRPA-based vaccine candidates.

*Indeed, the data in Figure 2 do suggest that rather high concentrations of these mAbs are required to see an inhibitory effect.*

Concentrations are comparable to those of published GIA data.

*3) Because the CyRPA is a conserved protein across several Plasmodium species as well as P. falciparum isolates, at least one reviewer would have liked to see data showing whether the inhibitory mAb inhibits strains other than 3D7.*

As requested by this reviewer, we have carried out additional in vitro GIAs with the *P. falciparum* strains K1, 7G8 and D6. Obtained results confirm the growth inhibitory activity of the anti-PfCyRPA c12 mAb and the anti-PfRH5 BS1.2 mAb, and their combination observed with strain 3D7. These data have been included as “Figure 2—figure supplement 1” in the manuscript.

*There is no mention of the specificity or the istotype of the control mAbs. Finally, it is important to show that mAbs specific for proteins other than RH5 do not have the same additive effect with anti-PfCyRPA. That control is missing here.*

This information has now been included in the manuscript.

To demonstrate that mAbs specific for proteins other than PfRH5 do not have the same additive effect with anti-PfCyRPA mAbs, we performed additional GIA experiments including an isotype-matched (IgG2a), malaria-unrelated control mAb (NR4.2).

Figure 2 has been updated to include data with mAb NR4.2.

*4) A key point for vaccine development is to show sterilizing immunity by targeting RH5 and CyRPA. In Figure 3, however, the authors do not show the sterilizing immunity that would be expected if the 2.5 mg CS1.2 and 2.5 mg c12 combination was tested. Was this performed? If not, why not? Since the authors have shown previously that c12 inhibits invasion, the only novelty here is the combination.*

While sterilizing immunity is an optimal effect of a vaccine, evidence generated so far for malaria indicates that this goal may be too ambitious for a single stage malaria vaccine. Classical passive transfer studies with serum antibodies from adults from malaria endemic areas by Cohen and others (e.g. Cohen et al., Nature 1961, 192:733–737 and Sabchareon et al., Am J Trop Med Hyg. 1991, 45(3):297-308), have shown that antibodies can reduce parasitemia considerably, but cannot confer sterile immunity. Maintenance of a basal parasitemia, which does not cause severe disease symptoms, is even considered by some experts in the field as desirable in order to confer booster effects.

In fact, we observe with anti-PfCyRPA and anti-PfRH5 antibodies a saturation effect without reaching sterile immunity. For investigating synergistic and additive effects, we had to use limited antibody concentrations.

Our results demonstrated that anti-PfCyRPA and anti-PfRH5 antibodies have an additive parasite growth inhibitory effect, justifying the combination of both antigens in a subunit vaccine. This is indeed highly relevant for vaccine development since the incorporation of the two vaccine components would help in preventing escape variant selection.

*5) The authors seem to have little experience with relatively low resolution structures, and their description of the refinement of the complex between Fab and PfCyRPA as "unsatisfactory" (subsection “Characterization of the epitope recognized by the parasite growth inhibitory anti-PfCyRPA mAb c12”, second paragraph) undermines confidence in the rest of the analysis, as one wonders whether the crystallographers involved (who are they?) really know how to look at a lower resolution map and how to avoid the pitfalls.*

The crystallographer in charge is M. Rudolph (trained with Ingrid Vetter, Ilme Schlichting, and Ian A. Wilson). The reviewer is correct that low resolution structures are not regular cases for us. We have tested various refinement strategies recommended by the PHENIX, CNS, BUSTER, and CCP4 developers. The refinement strategy that returned the best model, judged by R_free_ and the all-atom clash score (7.8) was similar to what was applied for other lower resolution structures, e.g. in Bradley et al. (2016; PMID 26725118) and Corbett et al. (2010; PMID 20723757): rigid body refinement of individual domains during molecular replacement, imposition of secondary structure restraints during positional refinement, and TLS and group ADP refinement for B-values. Additionally, the high resolution structures were used as reference models during refinement. The resulting PfCyRPA/c12 structure has much improved (~2% in R_free_), and we have removed the part “…and unsatisfactory refinement statistics of this structure…” in the manuscript. The now satisfactory refinement statistics have been included in the Table and are supported by a statistical analysis of the PDB using the PHENIX program:

phenix.r_factor_statistics 4.0 left_offset=0.5 right_offset=0.5 n_bins=10

Histogram of R_free_ for models in PDB at resolution 3.50-4.50 Å:

0.184-0.215: 23

0.215-0.247: 58

0.247-0.278: 158

0.278-0.310: 223

0.310-0.341: 200 ← R_free_ bin for the PfCyRPA/c12 complex

0.341-0.373: 90

0.373-0.404: 35

0.404-0.436: 10

0.436-0.467: 3

0.467-0.498: 3

*Since both structures have been refined at quite high resolution, and since even the CDR loops of the Fabs do not vary much from crystal form to crystal form, it seems quite odd that the complex is so poorly determined.*

This point is addressed below together with the elbow angles and the molecular replacement strategy.

*Specific points to consider are some red flags from data reduction statistics. Using the CC1/2 as the resolution cutoff criterion gives Rmeas of 59.5 in the outer bin (high for a low symmetry space group like P21) and I/sig(I) of 0.4 (and only 2.6 for the entire dataset).*

We have now used a resolution cut-off of 4 Å for the data, using the recommendation of A. Brunger’s group stated in DeLa Barre et al. (2006; PMID 16855310), where the resolution limit was chosen based on a sudden drop of σ_A_. At 4 Å resolution, σ_A_ for the PfCyRPA/c12 data drops below 0.8 (please see Figure 8; calculated with CCP4 program SIGMAA).

Author response image 2.**DOI:**
http://dx.doi.org/10.7554/eLife.20383.020

Reprocessing of the data to 4 Å resolution has improved overall I/σ from 2.6 to 3.3. I/σ in the outer shell has tripled from 0.4 to 1.2). R_meas_ has improved from 59% to 42%. The CC1/2 in the outer shell is now 71%, much higher than the previous cut-off of 30%. The new cut-off is stated in the Methods section, and the Table collecting the diffraction data and refinement statistics has been updated.

*Since the elbow varies a bit among Fab c12 crystal forms, did the authors use the variable and constant modules as separate MR search objects when determining the structure of the complex?*

We did use both, the variable and constant domains separately, as well as the entire Fab in separate molecular replacement searches together with the high-resolution PfCyRPA structure. Using the entire Fab yielded a log-likelihood gain of LLG = 1260, whereas the search with the individual Fv and Fc dimers had virtually identical LLG = 1270. This is reflected in the superposition of the molecular replacement solutions:

Author response image 3.**DOI:**
http://dx.doi.org/10.7554/eLife.20383.021

The molecular replacement solution using the Fab as a search model is shown as a tube (light chain magenta, heavy chain blue, PfCyRPA in green). The solution for the search using the F_v_ and F_c_ domains is shown as a ribbon (light chain orange, heavy chain green, PfCyRPA in white). This search reproduced the arrangement expected for the complete Fab. The elbow angle of the c12-Fab does not change appreciably in the complex with PfCyRPA, and any differences would have been corrected by rigid body refinement. (The two solutions that are shown in Figure 9 needed shifting along the polar axis for superposition).

Figure 10 shows electron density contoured at 1 rmsd for the linker connecting the Ig-domains of the light chain after molecular replacement (blue) and after initial refinement (black).

Author response image 4.**DOI:**
http://dx.doi.org/10.7554/eLife.20383.022

The linker (magenta) was not included in the refinement. As can be seen, there is weak electron density for the linker even after molecular replacement, which becomes connected after initial positional and group B-value refinement.

*Summary of required revisions:*

*1) Response to point (1), above, either with new data or – if a better structure of the c12:PfCyRPA complex can be achieved – by eliminating those experiments altogether, as a fully convincing structure will make them redundant.*

We addressed questions of point (1) with new IFA data, which are also included in the manuscript as supplemental figure (Figure 1—figure supplement 1).

*2) Explicit defense of the variant GIA assay, or use of a more standard, quantitative assay.*

We have adapted our protocol (based on a two cycles assay) to the one cycle GIA used in the publication of Chen et al. (published in 2014 in *eLife* – doi: 10.7554/*eLife*.04187). We have then repeated the in vitro GIA experiments with this modified protocol. Previous data have been replaced in the manuscript, and the new data have been included in the new version of Figure 2. Results with the more commonly used one cycle assay are comparable to those obtained before with the two cycle assay.

*3) Invasion inhibition assay with at least one more strain and inclusion, for both strains, of the stipulated control.*

Additional in vitro GIAs were carried out with *P. falciparum* strains K1, 7G8 and D6. Obtained results confirmed the growth inhibitory activity of anti-PfCyRPA c12 mAb and anti-PfRH5 BS1.2 mAb, and their combination, against strains other than 3D7. These data have been included as “Figure 2—figure supplement 1” in the manuscript.

An isotype-matched (IgG2a), malaria-unrelated control mAb has been included in all in vitro experiments presented in the updated manuscript.

*4) What combination of concentrations will give sterilizing immunity?*

Based on our findings we assume that sterilizing immunity may not be achievable with a single stage malaria vaccine. However, this can only be conclusively investigated by clinical trials.

Based on the received suggestions, we decided to investigate the in vivo effect of the combination of 2.5 mg anti-CyRPA c12 mAb + 2.5 mg anti-PfRH5 BS1.2 mAb. As expected, newly generated in vivo data confirmed the strong parasite growth inhibitory effect of anti-PfCyRPA and anti-PfRH5 antibodies administered in combinations:

In the control group parasitemia reached 19.8% on day 8 after mAb injection. Conversely, parasitemia in mice having received 2.5 mg c12 + 2.5 mg BS1.2 mAbs increased only marginally, reaching 0.99% on day 8 after mAb injection.

But as expected, despite the impressive results, sterilizing immunity could not be achieved also administering the high doses of anti-PfCyRPA and anti-PfRH5 antibodies (presence of parasites in the in peripheral blood of infected mice was confirmed by microscopic analysis of Giemsa-stained blood smears).

Author response image 5.NODscidIL2Rγnullmice received purified 2.5 mg anti-PfCyRPA c12 mAb and 2.5 mg anti-PfRH5 BS1.2 mAb by i.v. injections.PBS was used as negative control. Mice were then infected with *P. falciparum* 3D7 and parasitemia was monitored over five days. Values represent the mean parasitemia in human erythrocytes in peripheral blood.**DOI:**
http://dx.doi.org/10.7554/eLife.20383.023

*5) A better structure of the complex. Careful re-refinement of the structure seems in order, including careful re-examination of the initial MR solution.*

With the re-processed data, we are confident that the structure of the complex has improved. Also, the molecular replacement solution, judged by the same result using two different approaches, must be correct. Another strong argument for the correctness of the molecular replacement solution of the complex is the fact that when placing only the Fab, the remaining electron density, although uninterpretable, could be used as a search model for phasing the PfCyRPA data. We are convinced that this phasing strategy would not have been possible had the Fab been placed wrongly. This has been emphasized in the Methods section by stating: “Molecular replacement readily placed c12, and initial maps calculated from this partial solution revealed additional electron density at the tips of the Fab (Figure 5—figure supplement 1), which occupied a volume large enough to host PfCyRPA. […] The same arrangement of Fv and Fc was obtained when using the Fv and Fc parts of the Fab as separate search models, indicating that the elbow angle of c12 does not change upon binding to PfCyRPA.”

The Methods section has been expanded to make it clear that model building and refinement were done carefully:

“The final model of PfCyRPA was used to complete the molecular replacement phasing of the PfCyRPA/c12 complex (log-likelihood gain of >1260), density for which is shown in Figure 5—figure supplement 2. […] Strong geometric restraints were applied throughout model building of the PfCyRPA/c12 complex. All models were built with COOT {Emsley, 2010 #1736} and refined with PHENIX {Adams, 2010 #5885}.”

*It might be better to collect more data, as redundancy, even at the current resolution, will result in more accurately determined intensities.*

We have tested hundreds of crystals of the complex at the SLS, and the reported dataset is the best we could achieve. The usual resolutions of the datasets obtained is about 8 Å, and merging of data from several crystals in order to obtain higher multiplicity and better I/σ at least in the low resolution bins was unsuccessful, as judged by high merging R-values. Processing of datasets from isomorphous crystals was performed using the 4 Å dataset as a reference to avoid indexing ambiguities that would impede merging.

We would love to get better data, but currently we are unable to provide them. The biological question answered by the PfCyRPA/c12 complex structure was where the epitope of this neutralizing antibody on PfCyRPA was. This question can be answered at the current resolution of 4 Å. Subsequent mutagenesis experiments have confirmed the interpretation.

*Please provide some supplementary figures with maps and models, as was once standard, to restore confidence that the structure is correct. Show us the original MR-phased map in the region of the interface, together with the final model inside it, and then also the final 2Fo-Fc map, likewise with the model. One could still have problems even if these maps looked clear and well fit, but they will help.*

Figure 5—figure supplement 1 shows models and electron density after molecular replacement using only the Fab structure as the search model. The density for the missing PfCyRPA is visible. Figure 5—figure supplement 1 shows difference electron density for a loop region in PfCyRPA that was absent in the search model for molecular replacement. This loop could be traced with confidence, and the final model is also shown in the figure.

To further increase the confidence in the structure, we have added Figure 5—figure supplement 2 in the Supplementary Information to show the electron density of the PfCyRPA/c12 interface after molecular replacement, and after refinement, with the final model superimposed.